# Lamellar architectures in stiff biomaterials may not always be templates for enhancing toughness in composites

Michael A. Monn [1], Kaushik Vijaykumar[1], Sayaka Kochiyama[1] & Haneesh Kesari[1]*

The layered architecture of stiff biological materials often endows them with surprisingly high fracture toughness in spite of their brittle ceramic constituents. Understanding the link between organic–inorganic layered architectures and toughness could help to identify new ways to improve the toughness of biomimetic engineering composites. We study the cylindrically layered architecture found in the spicules of the marine sponge *Euplectella aspergillum*. We cut micrometer-size notches in the spicules and measure their initiation toughness and average crack growth resistance using flexural tests. We find that while the spicule's architecture provides toughness enhancements, these enhancements are relatively small compared to prototypically tough biological materials, like nacre. We investigate these modest toughness enhancements using computational fracture mechanics simulations.

---

[1] Brown University School of Engineering, 184 Hope St, Providence, RI 02912, USA. *email: haneesh_kesari@brown.edu

Despite being primarily composed of brittle ceramics, some stiff biological materials (SBMs), such as bones and shells, are remarkably tough[1–4]. For example, nacre—the iridescent material found in mollusk shells—is composed of >95% aragonite (a brittle, calcium carbonate mineral) by volume yet it has a specific fracture initiation toughness on par with nylon and some iron alloys[4]. A material's fracture initiation toughness quantifies its ability to prevent the growth of pre-existing cracks or flaws, and therefore be resistant to catastrophic failure. These tough SBMs are often heterogeneous and are composed of alternating layers of ceramic and organic materials (see Fig. 1). The arrangements of these layers, which we refer to as layered architectures, are thought to be responsible for the toughness enhancements observed in these SBMs[5]. There is currently considerable interest in understanding the connections between layered architectures and toughness enhancements in SBMs[5–8] because this understanding could aid in the development of new, tough engineering materials[4,9,10].

A number of SBMs, including nacre and bone, have served as models for the seminal research on understanding these connections[11,12]. Recently, the anchor spicules of the marine sponge *Euplectella aspergillum* (*Ea.*) have been added to this group of model SBMs[5,6,8,13]. The anchor spicules are hair-like fibers that attach *Ea.* to the soft sediment of the sea floor where it lives (see Fig. 2a, b)[14]. Each of the thousands of anchor spicules in a *Ea.* sponge is approximately 10 cm long and 50 μm in diameter. Viewed in cross-section, an anchor spicule consists of a solid cylindrical core surrounded by ≈25 concentric, cylindrical layers (see Figs. 1b and 2c)[14–16]. Both the core and the layers are composed of silica and adjacent silica layers are separated by a thin (≈5–10 nm[14]) organic interlayer. Similar cylindrical layered architectures have also been found in spicules from a number of related sponge species[17–20]. Images of spicules from other Hexactinellid species that are partially dissolved in alkali solution reveal that the silica layers also contain a fibrillar organic matrix similar to the interlayers[21–23]. Thus, this organic matrix serves both as a scaffold within the layers and a glue between them[24].

It is believed that this organic matrix acts as a template for cell-assisted silica mineralization during the spicule's growth process[21–23,25]. However, little is known about the growth process of *Ea.* spicules[25].

Many previous studies of *Ea.* anchor spicules suggest that like the layered architectures of nacre and other tough SBMs, the spicule's cylindrical layered architecture also enhances fracture toughness[5,6,8,13]. None of these studies, however, provide direct measurements of the *Ea.* spicule's fracture toughness, nor by other means do they quantify how much the architecture enhances the spicule's toughness compared to that of its constituent silica. These measurements and comparison are critical for determining whether the *Ea.* spicules should be used as a template for bioinspired materials with enhanced toughness.

In order to quantify fracture toughness enhancement provided by the *Ea.* spicule's architecture, its fracture toughness should be compared to that of a monolithic specimen consisting of the same biogenic silica. It has been shown previously that the spicules from a related sponge, *Tethya aurantia*, have a similar chemical composition and bonding structure to the *Ea.* spicules[18], but do not possess a layered architecture (see Fig. 2f)[26]. This makes the *Ta.* spicules a reasonable, if not ideal, choice as a control material for quantifying *Ea.* spicule's fracture toughness enhancement.

In this study, we first measure the *Ea.* and *Ta.* spicule's fracture toughness by cutting micrometer-size notches in the spicules using a focused ion beam and performing flexural tests on them. We then compare the fracture toughness of the *Ea.* and *Ta.* spicules and quantify the toughness enhancement provided by the *Ea.* spicule's architecture. We find that the toughness enhancement provided by the *Ea.* spicule's architecture is much smaller than that provided by architectures seen in prototypically tough SBMs, like nacre and bone. While these tough SBMs also possess layered architectures, the layers in these materials are flat rather than cylindrical/curved. For very short notches (less than 10% of the spicule's diameter), we observe that the *Ea.* spicule's architecture does provide up to a 10 fold increase in fracture initiation toughness. However, this enhancement is still relatively small

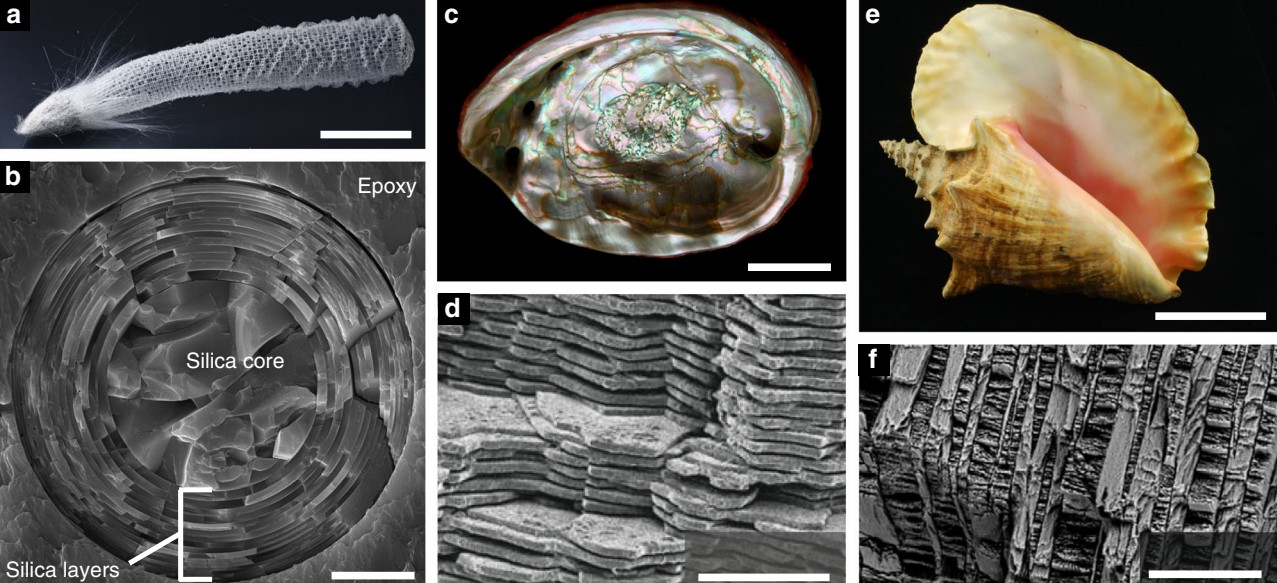

**Fig. 1 Examples of the layered architecture in SBMs. a** Skeleton of a *Euplectella aspergillum* sponge (modified from[16] copyright 2015, The National Academy of Sciences). **b** Concentric silica layers make up the cylindrical layered architecture of anchor spicules from *E. aspergillum* (modified from[16] copyright 2015, The National Academy of Sciences). **c** The iridescent shell of *Haliotis rufescens* (courtesy of John Varner). **d** The brick and mortar layered architecture of nacre consisting of staggered aragonite tablets (modified with permission from[65] copyright 2012, the Royal Society of Chemistry). **e** The shell of the queen conch (*Strombus gigas*) (courtesy of John Varner). **f** The crossed-layered architecture of the *S. gigas* shell, which consists of layers of aragonite (modified with permission from[66] copyright 2014, Elsevier). Scale bars: **a** ≈5 cm; **b** 10 μm; **c** ≈2 cm; **d** 10 μm; **e** ≈10 cm; **f** 100 μm.

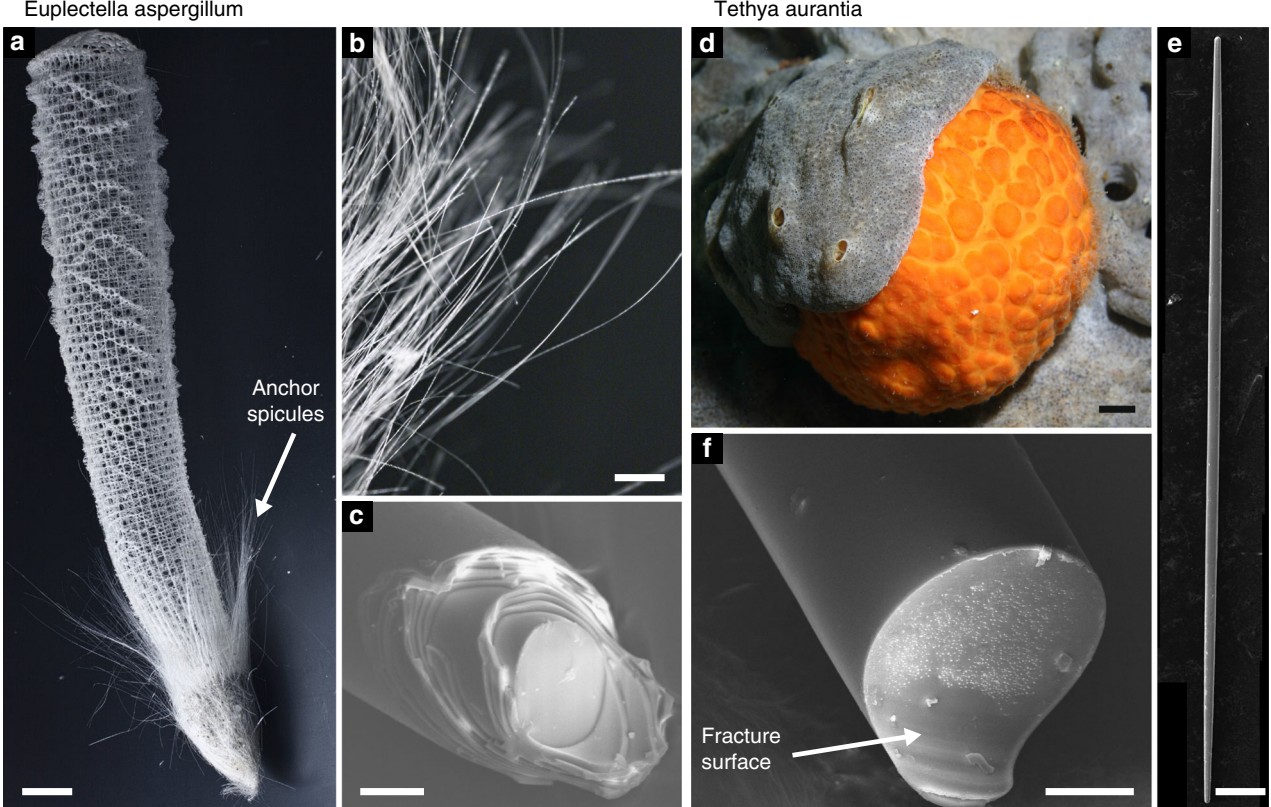

**Fig. 2 The skeletons and spicules of *Ea*. and *Ta*. sponges. a** The skeleton of the *Ea*. sponge (modified from[16] copyright, the National Academy of Sciences). **b** The anchor spicules that fasten the sponge to the seafloor (modified from[16] copyright, the National Academy of Sciences). **c** The broken end of an *Ea*. anchor spicule that was fractured in three-point bending showing its layered architecture (modified from[32] copyright 2017, Elsevier). **d** A *Ta*. sponge (image courtesy of Steve Lonhart/NOAA MBNMS). **e** A toothpick-like spicule found within the *Ta*. sponge (reproduced from[26] under the Creative Commons 4.0 BY license). **f** The exposed surface of a *Ta*. spicule that was fractured in the same way as the *Ea*. spicule shown in **c** showing that it does not contain a layered architecture (modified from[32] copyright 2017, Elsevier). Scale bars: **a** ≈2 cm; **b** 2 mm; **c** 25 μm; **d** ≈1 cm; **e** 125 μm; **f** 10 μm.

compared to that observed in nacre. Finally, we investigate the difference in the toughness enhancements provided by the cylindrical and flat layered architectures using computational mechanics simulations. We find that while crack arrest and re-nucleation appears to be the dominant toughening mechanism in the flat layered architecture, it does not manifest to the same extent in the cylindrical layered architecture. Thus, the curvature of the *Ea*. spicules' layers could fundamentally change the toughening mechanisms operating within them compared to SBMs with flat layered architectures.

## Results

**Recapitulation of the concept of fracture toughness**. Roughly speaking, fracture toughness—also known as crack growth resistance, $R$—is the amount of energy that a crack consumes to grow its area by a unit amount. If the energy consumed does not depend on the geometries of the crack, the crack increment, the specimen, and the specimen's architecture then $R$ is considered to be a material property. However, in SBMs the value of $R$ can depend on the length that the crack has grown, $\Delta a$ (see Fig. 3b, c). The value of $R$ when a crack first starts growing, $R(0)$, is known as the fracture initiation toughness. If the value of $R$ increases with $\Delta a$, then the material is said to have a rising $R$ curve (see Fig. 3c). In this case, as the crack grows the material becomes more resistant to crack growth. Several SBMs like nacre and bone[7,27–29] as well as in synthetic materials with architectures inspired by these SBMs (see Fig. 3c)[1] display rising $R$ curves. In these materials, the rise in $R$ is caused by toughening mechanisms that

become activated as a crack grows and interacts with the layered architecture.

We measured $R(0)$ and $\langle R \rangle$—i.e., the average value of $R$ (see Section Measurements of average crack growth resistance for details)—for the *Ea*. anchor spicules and the *Ta*. spicules (see Results section). By comparing $R(0)$ and $\langle R \rangle$ of the *Ea*. spicules to $R(0)$ and $\langle R \rangle$ of the *Ta*. spicules, we quantified the toughness enhancement provided by the *Ea*. spicule's architecture both at fracture initiation and during crack growth (see Section Comparison of toughness enhancements).

**Summary of experiments**. To measure $R(0)$ and $\langle R \rangle$, we performed flexural tests on 35 *Ea*. and 26 *Ta*. spicules using a configuration similar to that described by Jaya et al.[30,31]. We placed a spicule across a trench that was cut in a steel plate and ensured that its longitudinal axis was perpendicular to the trench edges (see Fig. 4a). We used trenches whose spans were nominally 600 to 800 μm and measured the span of each trench, $L$, from optical micrographs (see Table 1 for a summary and Supplementary Tables 2 and 3 for measurement details). We then glued the ends of the spicule to the steel plate so that only the section suspended over the trench remained exposed.

The spicule specimen's undeformed configuration can be described using the orthonormal set of Cartesian basis vectors $\{\hat{e}_1, \hat{e}_2, \hat{e}_3\}$ (Fig. 4a, b, d), which correspond to the Cartesian coordinates $\{x_1, x_2, x_3\}$. The origin of this coordinate system, denoted as $\mathcal{O}$, is located at the point on the spicule's central, longitudinal axis directly above the left trench edge (see Fig. 4a).

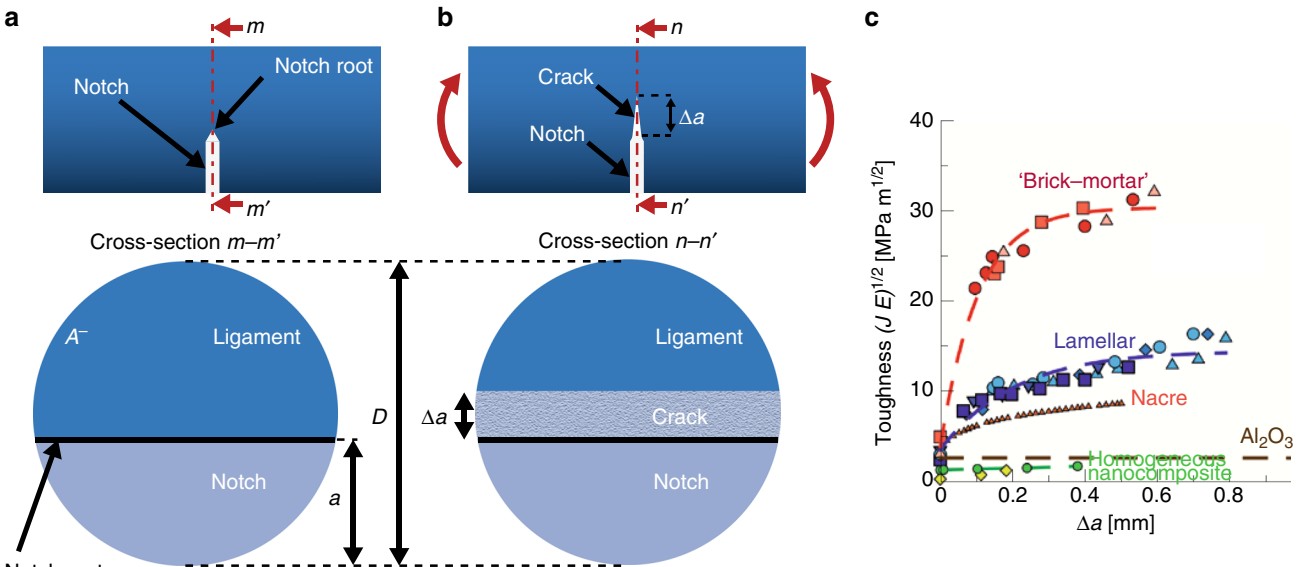

**Fig. 3 Fracture toughness and crack growth from a notch. a** Geometry of a fracture test specimen with a notch cut in it. A cross-sectional view shows the notch length, $a$, and ligament area, $A^-$. **b** When loaded in flexure, a crack grows from the notch root. The crack length, $\Delta a$ is shown in the corresponding cross-sectional view. **c** Crack growth resistance curves of nacre, nacre-inspired composites made of aluminum oxide tablets, and monolithic aluminum oxide obtained by[1] and[9]. The crack growth resistance is given here in terms of the $J$-integral (modified with permission from[1] copyright Nature Publishing Group).

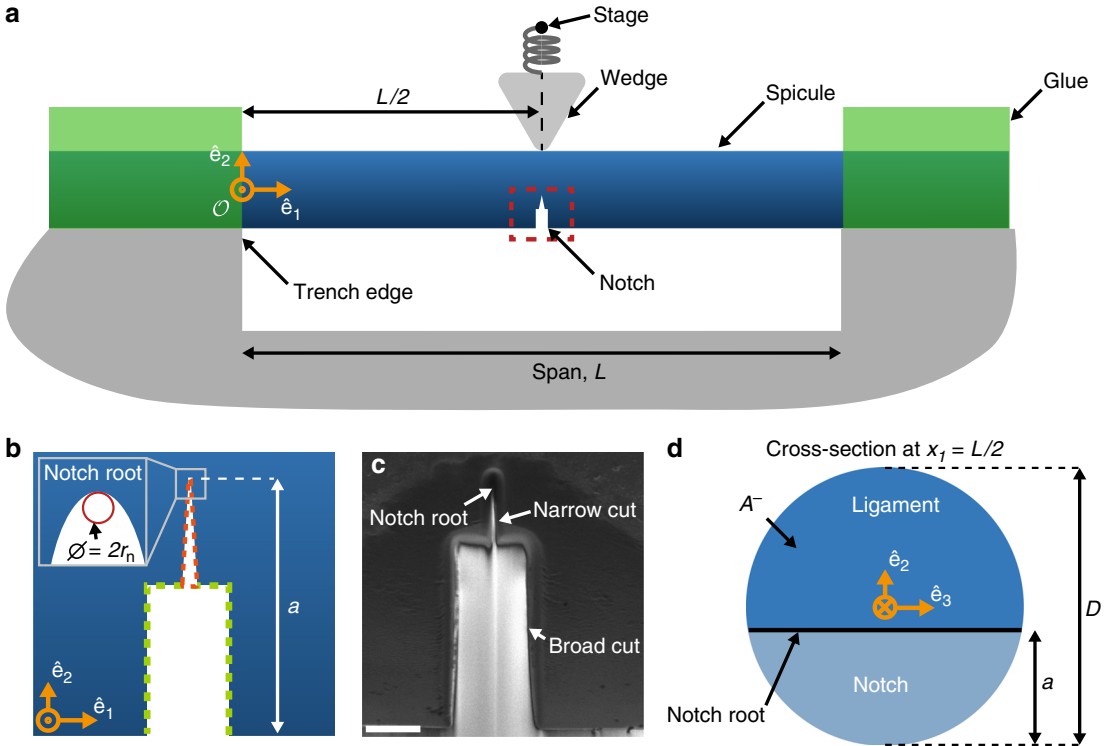

**Fig. 4 Notched spicule specimen geometry. a** A schematic of the test configuration. The stage is denoted by a black dot. **b** A magnified view of the region shown in the red rectangle in **a** showing the notch geometry. The broad cut that was made using a high accelerating current is marked in green and the narrow cut that was made using the low accelerating current is marked in orange. **c** A scanning electron micrograph of the notch cut in a representative *Ea.* spicule. Scale bar is 2.5 μm. **d** A schematic of the spicule's cross-section at $x_1 = L/2$ after notching. The notch root is straight and parallel to $\hat{e}_3$. The notch length is $a$. The remaining ligament has a cross sectional area $A^-$.

We cut a notch through part of the spicule's cross-section located mid way across the trench (i.e., at $x_1 = L/2$) using a focused ion beam (FIB) (see Fig. 4a, b and Section Spicule notching procedure). A representative micrograph of a notched spicule is shown in Fig. 4c. Figure 4d depicts a schematic representation of the spicule's cross-section at $x_1 = L/2$ in which the notched region is shown in light blue and the remaining ligament is shown in dark blue. We refer to the apex or tip of the notch as the notch root. The notch root is a straight line segment that is parallel to $\hat{e}_3$, see Fig. 4d). After cutting the notch, we

**Table 1 Summary of specimen geometry for *Ea.* and *Ta.* spicules.**

| Species | No. specimens | $L$ ($\mu$m) | $D$ ($\mu$m) | $a$ ($\mu$m) |
|---|---|---|---|---|
| *E. aspergillum* | 35 | 799.25 ± 6.35 | 41.42 ± 2.16 | 12.92 ± 1.30 |
| *T. aurantia* | 26 | 725.86 ± 18.36 | 32.28 ± 1.07 | 6.96 ± 0.80 |

Values for diameter ($D$), span ($L$) and notch length ($a$) listed as mean ± standard error of measurement (see Fig. 4).
[a]Supplementary Tables 2 and 3 for details of the geometry of individual *Ea.* and *Ta.* specimens.

imaged the spicule using the FIB and measured the diameter of the spicule's cross-section at $x_1 = L/2$, $D$, and the notch length, $a$, from the micrographs (see Table 1 for a summary and Supplementary Tables 2 and 3 for measurement details).

We positioned the spicule underneath a steel wedge so that the apex of the wedge was located at $x_1 = L/2$ on the opposite side of the spicule from the notch (see Fig. 4a). We used a motorized translation stage to push the wedge into the spicule in 1 μm displacement increments at a rate of 1 μm s$^{-1}$. The displacement of the translation stage is $-w_s \hat{\mathbf{e}}_2$ (see Fig. 5a). We also measured the displacement of the spicule's cross-section beneath the wedge, $w_0$ (see Fig. 5a). The device used to perform the flexural tests is described in detail in[32,33].

The wedge was attached to a cantilever whose stiffness was measured before the test. We measured the deflection of the cantilever using a fiber optic displacement sensor. The force acting on the spicule is $-F\hat{\mathbf{e}}_2$. We computed $F$ using the cantilever's stiffness and the measured deflection[32]. Since spicule specimens with longer notches require less force to fracture, we used cantilevers with different stiffnesses depending on the spicule diameter and notch length. The cantilever stiffnesses in our experiments ranged from 88 to 9100 Nm$^{-1}$.

**Force-displacement responses of notched spicules**. Representative $F$-$w_0$ data for an *Ea.* and *Ta.* spicule are shown as dark gray points in Fig. 5c, d, respectively. We observed that $F$ first increases with $w_0$ up to a value of $F_c$, at which point there is an abrupt drop in force. We interpret this abrupt drop in force to be the point at which a crack starts growing from the notch root. This event is commonly referred to as pop-in[30,34]. We denote the displacement corresponding to $F_c$ as $w_c$. The point ($w_c$, $F_c$) is shown as a red square in Fig. 5c, d and the values of $w_c$ and $F_c$ for each specimen are given in Supplementary Tables 2 and 3.

As we continued to load the spicule after pop-in, the crack propagated across the spicule's cross-section until it completely cleaved the spicule into two pieces. Finally, we unloaded the spicule by moving the stage away from the spicule (i.e., in the $\hat{\mathbf{e}}_2$ direction) in 1 μm displacement increments at a rate of 1 μm s$^{-1}$. The $F$-$w_0$ data obtained during unloading are shown as light gray points in Fig. 5c, d.

After the spicule was completely unloaded we dissolved the adhesive on its ends and obtained two separate pieces, which we collected for additional imaging (see Section Fractography).

**Fractography**. After testing the *Ea.* and *Ta.* spicule specimens, we imaged their fracture surfaces using a scanning electron microscope (Fig. 6a, b). In all *Ea.* and *Ta.* specimens, failure appears to occur via a single crack that originates at the notch root. The existence of a single dominant crack is a prerequisite to computing the spicules' average crack growth resistance, which we do in Section Measurements of average crack growth resistance.

The fracture surfaces of both the *Ea.* and *Ta.* spicules appear to be relatively featureless. In the case of the *Ea.* spicules, this contrasts with the fracture surfaces observed in other SBMs with layered architectures, like nacre and conch shell (see Fig. 1d, f). In these other SBMs the fracture surfaces appear very rough. This

roughness is thought to be a signature of the crack arrest and re-nucleation toughening mechanism that occurs when a crack reaches an interface between adjacent layers[10,35–37]. As such, the relatively smooth fracture surface of the *Ea.* spicules suggests that they may not possess the same toughening mechanism(s) associated with these other SBMs.

The fracture surfaces of both the *Ea.* and *Ta.* spicules have a cusp feature adjacent to where the load is applied (see e.g., Fig. 6a, b). This cusp is characteristic of the three point bending configuration[38] and is a consequence of the compressive stresses caused by the wedge inducing local mixed-mode fracture conditions, which cause the crack to change direction[39]. The cusp is not a result of a toughening mechanism caused by the *Ea.* spicules architecture since it also appears in the *Ta.* spicules, which lack the layered architecture.

**Measurements of fracture initiation toughness**. During pop-in, a crack grows from the notch root in the transverse direction, across the spicule's cross-section (i.e., in the $\hat{\mathbf{e}}_2$ direction, see Fig. 3b). We assume that the crack front is straight and parallel to the notch root (i.e., parallel to $\hat{\mathbf{e}}_3$) and denote the crack length as $\Delta a$. The energy release rate $G$ is given by

$$G(\Delta a; w_s) = -\frac{1}{2\sqrt{(a + \Delta a)(D - (a + \Delta a))}} \frac{d\Pi(\Delta a; w_s)}{d\Delta a},$$ (1)

where $\Pi(\Delta a; w_s)$ is the system's potential energy when the crack's length is $\Delta a$ and the applied displacement is $w_s$. For a derivation of Eq. (1), see Supplementary Note 1. It follows from Irwin's analysis of Griffith's theory of fracture that the necessary condition for crack growth is $G(\Delta a, w_s) \geq R(\Delta a)$, where $R(\Delta a)$ is the material's crack growth resistance[39]. We assume that crack growth first occurs when $G(0, w_s) = R(0)$ and $w_s$ is the applied displacement at pop-in. Thus, the fracture initiation toughness $R(0)$ is given by

$$R(0) = -\frac{1}{2\sqrt{a(D-a)}} \frac{d\Pi(\Delta a; w_s)}{d\Delta a}\bigg|_{\Delta a = 0},$$ (2)

where $w_s$ is the applied displacement at pop-in.

For each *Ea.* and *Ta.* spicule that we mechanically tested, we measured $R(0)$ by computing the derivative in Eq. (2) using a computational mechanics model. This model is described in detail in Supplementary Note 2. In the computational mechanics model we consider both the *Ea.* and *Ta.* spicules to be made of a homogeneous, linear elastic material. In reality, the *Ea.* spicules contain layers and are therefore not homogeneous. Therefore, the values of $R(0)$ that we obtain for the *Ea.* spicules should be considered to be an *effective* fracture initiation toughness.

The accurate estimation of $R(0)$ using Eq. (2) is predicated on the assumption that the notch behaves like a sharp crack (i.e., the radius of curvature of the notch root is vanishingly small). It has been shown that the FIB cutting technique can produce notch root radii that are small enough to act like sharp cracks[40]. This is supported by additional work[41] showing that if the notch root radius is less than twice the smallest microstructural length scale,

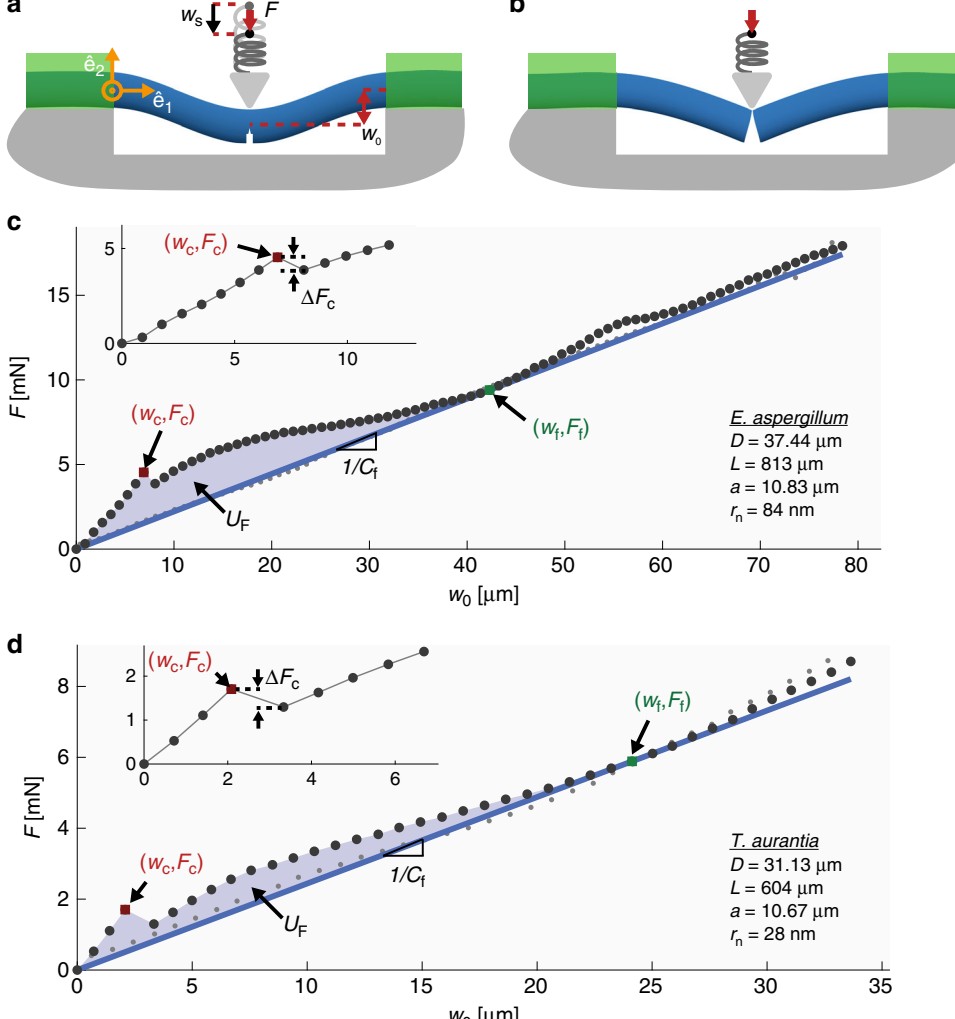

**Fig. 5 Data obtained from the bending tests on *Ea*. and *Ta*. spicules. a** The wedge is used to apply a force $-F\hat{\mathbf{e}}_2$. The displacement of the stage is $-w_s\hat{\mathbf{e}}_2$ and the corresponding displacement of the spicule's cross-section at $x_1 = L/2$ is $-w_0\hat{\mathbf{e}}_2$. The stage is denoted by a black dot. The glue (shown in green) prevents the ends of the spicule from rotating or moving relative to the plate to which it is attached. **b** After the specimen has failed completely, it resembles two cantilevers. **c** The $F$-$w_0$ response of a representative *Ea*. spicule. **d** The $F$-$w_0$ response of a representative *Ta*. spicule. The $F$-$w_0$ data obtained during loading and unloading are shown as dark gray points and light gray points, respectively. Crack initiation (i.e., pop-in) is marked as a red square and has a force and displacement of $F_c$ and $w_c$. The insets in **c** and **d** show a magnified view of the $F$-$w_0$ response leading up to pop-in and the drop in force during pop-in, $\Delta F_c$. The point of complete failure is marked with a green square and has a force and displacement of $F_f$ and $w_f$. The blue line with slope $1/C_f$ is the line that passes through the origin and the point $(w_f, F_f)$. The shaded region marked $U_F$ in **c** and **d** is the area enclosed between the $F$-$w_0$ data and this line.

then the measured value of $R(0)$ becomes insensitive to the notch root geometry. The *Ea*. spicules' layers are composed of silica nanoparticles that are approximately 100 nm in diameter[15]. We take the size of these nanoparticles to be the smallest microstructural length scale present in the *Ea*. spicules. We assume that the *Ta*. spicules have a similar smallest microstructural length scale. This assumption is supported by atomic force microscopy of the *Ta*. spicules[42]. Therefore, in order for the value of $R(0)$ to be insensitive to the notch root geometry, the radius of curvature of the notch root, $r_n$, should be less than 200 nm (see Fig. 4b inset).

We measured $r_n$ for each specimen from scanning electron micrographs by manually selecting three points along the profile of the notch root and fitting a circle to these points (see Supplementary Tables 2 and 3). The mean value ± standard error of $r_n$ for the 35 *Ea*. spicules and 26 *Ta*. spicules was 112 ± 14 nm and 149 ± 15 nm, respectively. We identified 4 *Ea*. spicule

specimens and 5 *Ta*. spicule specimens for which $r_n$ exceeded 200 nm, and consequently we did not compute $R(0)$ for these specimens. Additionally, there were 9 *Ea*. spicule specimens for which we were unable to reliably identify the pop-in event by inspecting the $F$-$w_0$ response, and therefore could not obtain $w_c$. Thus, we computed $R(0)$ for 22 *Ea*. spicules and 21 *Ta*. spicules (see Fig. 7a).

We measured $R(0)$ for the *Ta*. spicules to be $3.76 \pm 0.49\,\mathrm{J\,m^{-2}}$ (mean ± standard error, $N = 21$) and $R(0)$ for the *Ea*. spicules to be $7.15 \pm 1.83\,\mathrm{J\,m^{-2}}$ (mean ± standard error, $N = 22$). The measurements of $R(0)$ for each spicule are shown in Fig. 7a and in Supplementary Tables 2 and 3. Overall, these values are similar to those expected for glass and other brittle ceramic materials (see Fig. 7a). In the case of the *Ta*. spicules, the value of $R(0)$ appeared relatively constant regardless of the dimensionless notch length, $\alpha = a/D$. For values of $\alpha > 0.1$, the fracture initiation toughness of the *Ea*. spicules was also relatively constant. However, the *Ea*.

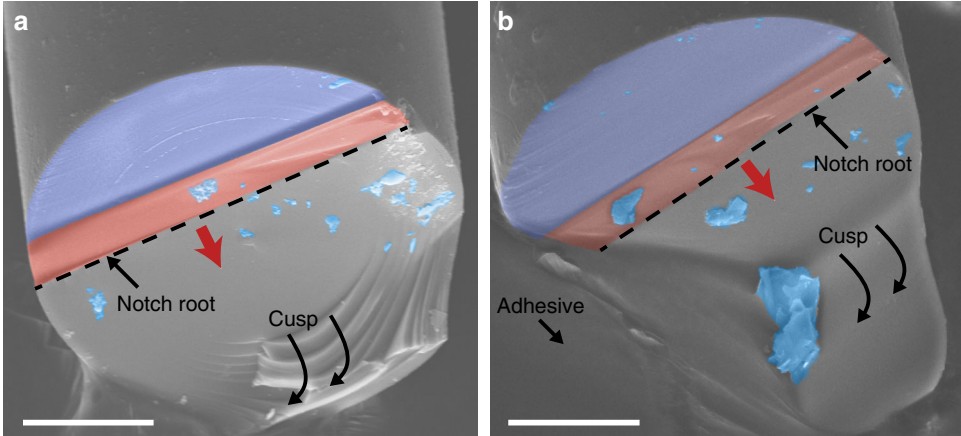

**Fig. 6 Micrographs of fractured *Ea*. and *Ta*. spicules.** Representative *Ea*. and *Ta*. spicules are shown in **a** and **b**, respectively. False color is used to mark important features. The purple region in both **a** and **b** corresponds to the coarse (higher current) FIB cut discussed in Section Spicule notching procedure. The red region in **a** and **b** corresponds to the fine (lower current) FIB cut. The blue regions correspond to debris that we assume collected on the fracture surfaces during specimen preparation. The red arrow denotes the direction of crack growth. The cusp feature in **a** and **b** is a feature of the three-point bending test configuration that appears for both the *Ea*. and *Ta*. spicules. In **b**, part of this feature is occluded by the adhesive used to mount the spicule to the aluminum stub. Scale bars are 10 $\mu m$.

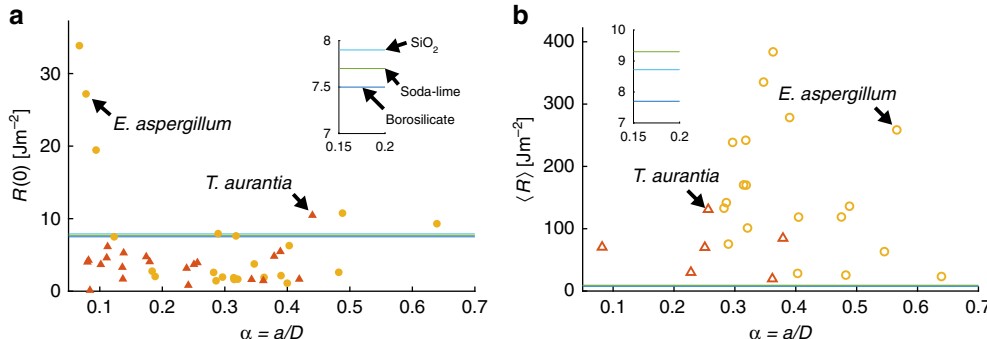

**Fig. 7 Crack growth resistance of *Ea*. and *Ta*. spicules. a** Fracture initiation toughness, $R(0)$, of 22 *Ea*. spicules (yellow circles) and 21 *Ta*. spicules (red triangles) versus dimensionless notch length, $\alpha = a/D$. Reference lines are shown corresponding to the fracture initiation toughnesses of three types of synthetic glass measured by[67]. The inset shows a closer view of these three values. **b** Average crack growth resistance, $\langle R \rangle$, of 19 *Ea*. spicules (hollow yellow circles) and 6 *Ta*. spicules (hollow red triangles) versus $\alpha$. Reference lines are given that correspond to the average crack growth resistance of the three types of synthetic glass shown in **a**. We computed the value of $\langle R \rangle$ from the measurements of the work of fracture provided by[67]. The inset in **b** shows a closer view of these three values.

spicules displayed a sharp increase in $R(0)$ for $\alpha < 0.1$ (see Fig. 7a). The largest value of $R(0)$ that we measured was 33.84 $\mathrm{Jm^{-2}}$ for an *Ea*. spicule specimen with $\alpha = 0.07$. The increase in fracture initiation toughness for small values of $\alpha$ suggests that the *Ea*. spicule's architecture is increasing its effective fracture initiation toughness for small flaws or cracks.

We offer the following qualitative explanation for this observed increase in $R(0)$. Recall that the *Ea*. spicule's cross-section consists of ≈25 cylindrical layers surrounding a large, monolithic core. In a previous work, it was shown that the ratio of the core's diameter to the spicule's diameter is relatively constant with a mean value of 0.41 and a standard deviation of 0.07 ($N = 116$)[16]. From this it can be shown that the value of $\alpha$ at which the notch reaches the core of the spicule is approximately 0.3. Thus, for $\alpha < 0.3$ a crack growing from the notch will initially propagate perpendicular to some of the interlayers. It has been shown that when a crack propagates perpendicular to the spatial variation of material properties in a structure consisting of flat layers of a stiff material that are separated by thin, compliant or weak interlayers, the structure will display enhanced fracture initiation toughness compared to the bulk material[8,43,44]. The toughness enhancement

is a result of the energy release rate decreasing when a crack impinges on a compliant or weak interlayer[8,43]. This mechanism has been observed experimentally in bioinspired composites[45,46] and previous works have speculated that it may also be operating in the *Ea*. spicules[8,43]. Our observation that $R(0)$ increases with decreasing $\alpha$ is consistent with the predictions of these models[8,43,44]. For specimens with $\alpha > 0.3$, the notch reaches the silica core of the spicule where there are no interlayers, and the interlayers between the silica layers outside the core are not perpendicular to the initial direction of crack growth. Therefore, in these specimens the spicule would not be able to benefit from the above-mentioned toughness enhancement mechanism. In agreement with this observation, in the *Ea*. spicules with $\alpha > 0.3$, the values of $R(0)$ are similar to those that we measured in the *Ta*. spicules (see Fig. 7a).

**Measurements of average crack growth resistance.** In our experiment the average crack growth resistance, $\langle R \rangle$ is defined as

$$\langle R \rangle = \frac{1}{D-a} \int_0^{D-a} R(\Delta a)\, \mathrm{d}\Delta a. \quad (3)$$

We measured $\langle R \rangle$ using the work of fracture method[47,48]. Specifically, it can be shown that

$$\langle R \rangle = 2\gamma_{\text{WOF}}, \tag{4}$$

where $\gamma_{\text{WOF}}$ is called the work of fracture[48]. The work of fracture is obtained by fracturing a specimen, measuring the total energy that is consumed by the fracture process, $U_{\text{F}}$, and dividing that by the total new surface area created[47]. Because the spicule specimens are cleaved into two pieces by a single crack emanating from the notch root (see Section Fractography), we take the total new surface area created to be twice the cross-sectional area of a specimen's ligament before any crack growth has occurred, $A^-$ (see Fig. 4d). Therefore, $\langle R \rangle$ is given by

$$\langle R \rangle = U_{\text{F}}/A^-. \tag{5}$$

In order to compute $\langle R \rangle$ using Eq. (5), the crack must grow in a stable manner throughout the test[49,50]. Unstable crack growth events appear as discontinuities in the $F$-$w_0$ response. The pop-in (marked by a red square in Fig. 5c, d), is an example of such an event. Consequently, $\langle R \rangle$ computed using Eq. (5) from the $F$-$w_0$ data will be an overestimate of the actual value of $\langle R \rangle$. Aside from the pop-in event, however, the $F$-$w_0$ curves shown in Fig. 5 are continuous. For that reason, we believe that in our experiments the cracks grow in a predominantly stable manner. Previous studies have attempted to compute the spicule's work of fracture from their $F$-$w_0$ response[6,13]. However, the data presented in these previous studies suggests that the crack growth was not stable and therefore the data in these studies cannot be used to obtain accurate estimates of $\langle R \rangle$.

We used the magnitude of the drop in $F$ at pop-in as a criterion for determining which spicule specimens could be used to estimate $\langle R \rangle$. Specifically, we measured the drop in force during pop-in, $\Delta F_c$ (see Fig. 5c, d inset), and compared it to $F_c$. We considered a test to have predominantly stable crack growth if $\Delta F_c \leq 0.15 F_c$. We identified 16 $Ea.$ and 20 $Ta.$ spicules for which $\Delta F_c > 0.15 F_c$ and consequently we could not compute $\langle R \rangle$ for these specimens.

If the crack grows in a predominantly stable manner then the change in kinetic energy of the spicule during the test is negligible and it can be shown that

$$U_{\text{F}} + U_{\text{ef}} = \int_0^{w_{\text{f}}} F \, dw_0, \tag{6}$$

where $w_{\text{f}}$ is the value of $w_0$ at which the crack has just cleaved the specimen into two pieces and $U_{\text{ef}}$ is the elastic energy of the spicule in the completely failed state. The integral on the right hand side of Eq. (6) is the work done on the spicule until it has completely failed. In order to evaluate this integral we must identify the $(w_0, F)$ point at which the crack had just cleaved the spicule into two pieces.

We assume that if at any point during the test we were to unload the spicule, $F$ would decrease linearly with $w_0$ until the values of both reached zero. Our rationale for this assumption is that the spicule behaves in a linear elastic fashion when there are no dissipative processes operating (e.g., crack growth). This means that the elastic unloading compliance of the spicule at a point $(w_0, F)$ is given by $C = w_0/F$. Since crack growth invariably increases the elastic unloading compliance of a structure, we know that $C$ should be greatest when the crack has completely cleaved the spicule into two pieces. Thus, we define $w_{\text{f}}$ and the corresponding force, $F_{\text{f}}$, at which the spicule has just been cleaved into two pieces to be the $(w_0, F)$ point for which $C$ is maximum. The point $(w_{\text{f}}, F_{\text{f}})$ is shown in Fig. 5c, d as a green square. The line with slope $1/C_{\text{f}}$ that passes through both this point and the origin is also shown in Fig. 5c, d in blue. The values of $w_{\text{f}}$ and $F_{\text{f}}$ for each specimen are given in Supplementary Tables 2 and 3.

When our specimen has completely failed, the elastic energy is given by $U_{\text{ef}} = w_{\text{f}} F_{\text{f}}/2$. We can therefore simplify Eqs. (5) and (6) to get

$$\langle R \rangle = \frac{\int_0^{w_{\text{f}}} F dw_0 - w_{\text{f}} F_{\text{f}}/2}{A^-}. \tag{7}$$

For a specimen with a circular cross-section containing a notch of length $a$ and crack of length $\Delta a$, as shown in Fig. 3b, the area of the intact portion of the specimen's cross-section is

$$A(\Delta a) = \frac{\pi D^2}{4} + \frac{1}{2}(D - 2(a + \Delta a))\sqrt{(a + \Delta a)(D - (a + \Delta a))} - \frac{D^2}{4}\cos^{-1}\left(1 - \frac{2(a + \Delta a)}{D}\right). \tag{8}$$

In Eq. (8) we assume that both the notch root and crack front are straight line segments that are parallel to $\hat{\mathbf{e}}_3$. We computed $A^- = A(0)$ from Eq. (8) using the $a$ and $D$ that we measured from scanning electron micrographs taken before the flexural tests (see Section Results).

For 19 $Ea.$ and 6 $Ta.$ spicule specimens we computed $\langle R \rangle$ from Eq. (7) using trapezoidal integration of the $F$-$w_0$ data up to the point $(w_{\text{f}}, F_{\text{f}})$. We found $\langle R \rangle$ to be $160.12 \pm 23.99$ Jm$^{-2}$ (mean $\pm$ standard error, $N = 19$) and $67.56 \pm 16.37$ Jm$^{-2}$ (mean $\pm$ standard error, $N = 6$) for the $Ea.$ and $Ta.$ spicules, respectively. The measurements of $\langle R \rangle$ for each spicule are shown in Fig. 7b and in Supplementary Tables 2 and 3.

**Comparison of toughness enhancements**. We found that the average crack growth resistance of the $Ea.$ spicules was higher than that of the $Ta.$ spicules. Specifically, computing the ratio $\langle R \rangle^{\text{(EA)}}/\langle R \rangle^{\text{(TA)}}$ shows that the architecture increases $\langle R \rangle$ by a factor of 2.37. While this ratio is evidence that the $Ea.$ spicule's architecture does enhance toughness, we should put this enhancement in the context of enhancements observed in other SBMs. We denote the crack growth resistances of an architectured material and its corresponding homogeneous ceramic constituent with the superscripts (arch) and (hom), respectively. By computing $\langle R \rangle^{\text{(arch)}}/\langle R \rangle^{\text{(hom)}}$ for nacre, bone, antler and conch shell using work of fracture data available from literature (see Fig. 8b and Supplementary Table 1), we see that the $\langle R \rangle$ enhancement in the $Ea.$ spicules is quite small. For example, $\langle R \rangle^{\text{(arch)}}/\langle R \rangle^{\text{(hom)}}$ for the conch shell exceeds 1000, several hundred times larger than the $\langle R \rangle$ enhancement in the $Ea.$ spicules.

By computing the enhancement in $R(0)$ using the same procedure, we see that $R(0)^{\text{(arch)}}/R(0)^{\text{(hom)}}$ can be as large as $\approx 200$ in these other biological materials (see Fig. 8a). In contrast, we measured $R(0)^{\text{(arch)}}/R(0)^{\text{(hom)}}$ to be only 1.90 for the $Ea.$ spicules on average. Even when considering the relatively large increase in $R(0)$ for short notches (see Section Measurements of fracture initiation toughness for details), the maximum enhancement that we observed is on the order of 10. While this increase is similar to those observed in conch and antler it still is quite small compared to nacre—often considered the archetype for tough biological materials. Thus, while the $Ea.$ spicules share a common architectural motif with many tough SBMs, our measurements suggest that these seemingly similar architectures do not provide comparable enhancements to either the fracture initiation toughness or average crack growth resistance.

Other studies have compared the mechanical behaviors of spicules with layered architecture to synthetic glass fibers[6,13,18,19,51], despite the spicules having a lower elastic modulus and a different chemical composition[18,52,53]. Unlike synthetic glass, the $Ea.$ spicules are composed of hydrated silica that is precipitated onto

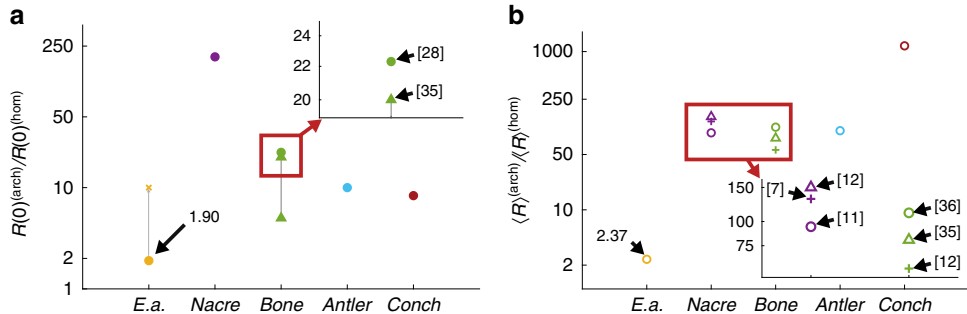

**Fig. 8 Comparison of toughness enhancements in the *Ea.* spicules and other SBMs with layered architectures. a** The initiation toughness enhancement metric $R(0)^{(\text{arch})}/R(0)^{(\text{hom})}$ of the *Ea.* spicules and other SBMs with layered architectures. We used the value $R(0)^{(\text{hom})} = 3$ Jm$^{-2}$ of calcite as the control material for nacre and conch[68], and $R(0)^{(\text{hom})} = 10$ Jm$^{-2}$ of hydroxyapatite[68] as the control material for bone and antler. For *Ea.*, the solid circle represents the average and the cross represents the approximate value of the maximum enhancement which we observed for very short notches. **b** The average toughness enhancement metric $\langle R \rangle^{(\text{arch})}/\langle R \rangle^{(\text{hom})}$ of the *Ea.* spicules and other SBMs with layered architectures. We used the value $\langle R \rangle^{(\text{hom})} = 22$ Jm$^{-2}$ of limestone as the control material for nacre and conch[69], and $\langle R \rangle^{(\text{hom})} = 140$ Jm$^{-2}$ of hydroxyapatite[70] as the control material for bone and antler. In both **a** and **b**, the values of $R(0)$ and $\langle R \rangle$ for nacre, bone, antler, and conch (*S. gigas*) were obtained from the literature (see Supplementary Table 1). The multiple values for bone in **a** and for nacre and bone in **b** correspond to experiments performed by different research groups. The two green triangles connected by a line in **a** correspond to the range of values reported by[35]. The references from which the data points were obtained are annotated in the insets. Note that in both **a** and **b** the toughness enhancement metrics are plotted using a logarithmic scale.

a proteinaceous scaffold[54]. The effect of this scaffold on the mechanical properties of the spicule's silica is not fully understood. It would therefore not be possible to isolate the effect of the spicule's layered architecture on its toughness properties by comparing them to synthetic glass fibers. The *Ea.* spicules should instead be compared to a specimen composed of the same biogenic silica but which is monolithic. An ideal choice for this homogeneous control material would be a section of the solid silica core of the *Ea.* spicules. However, so far we have not successfully obtained a large enough section of the *Ea.* spicule core to perform fracture tests. Therefore, we chose the *Ta.* spicules as what we believe to be the next best alternative.

## Discussion

Toughness enhancements in SBMs are often caused by multiple mechanisms that are triggered as a crack interacts with the features of the material's architecture[12,27,55]. To understand the difference between the toughening mechanisms operating in the *Ea.* spicules and in SBMs like nacre we performed virtual experiments using a regularized variational fracture (RVF) method (see Supplementary Note 3 for details). These virtual experiments allowed us to predict the mechanical behaviors of and crack paths in materials with layered architectures. Specifically, we used the RVF method to simulate crack growth in a material with flat/planar layers, like those which appear in nacre, and a material with cylindrical layers, like in the *Ea.* spicules.

Our model material with a planar layered architecture consists of a notched beam with a rectangular cross-section whose width $W = 1$ mm, thickness $H = 1$ mm and length $L = 5$ mm. The beam is composed of two layers separated by a thin interlayer (see Fig. 9a). Similarly, our model material with a cylindrical layered architecture consists of a notched beam with a cylindrical cross-section composed of two layers separated by a cylindrical interlayer (see Fig. 9b). We constrained the cylindrical beam to have the same volume as the rectangular beam and therefore its diameter $D = 1.12838$ mm and length $L = 5$ mm. The notch length $a$ in both beams is 0.2 mm.

In both beams, the interlayer has a thickness $t = 0.1$ mm and is located a distance $b = 0.2$ mm from the notch root. The Young's modulus $E = 20.8$ GPa and Poissons ratio $\nu = 0.3$ are the same in both the layers and the interlayer. However, the fracture toughness is $G_b = 500$ Jm$^{-2}$ in the layers and $G_I = 0.5$ Jm$^{-2}$ in the

interlayer. We loaded the beams in three-point bending by applying a displacement $w_0$ shown in Fig. 9.

In Fig. 9c we show the load-displacement ($F$-$w_0$) response of the planar layered beam as well as that of a beam with the same geometry but without the interlayer. The monolithic beam exhibits typical brittle behavior with crack initiation occurring at $F \approx 25$ N leading to a large drop in $F$, which corresponds to abrupt crack growth, followed by a decrease in $F$ until complete failure. The planar layered beam shows an almost identical response until the peak load is reached and first load drop occurs. After the first load drop, however, the load again increases before a second load drop occurs leading to complete failure. The crack path for the planar layered beam at $w_0 = 0.16$ mm is shown in Fig. 9d. By examining the crack path at different values of $w_0$, we observe three stages of crack growth. First, a crack initiates at the peak load and grows until it reaches the interlayer. This corresponds to the first load drop from $F \approx 25$ N to $F \approx 9$ N. Once the crack reaches the interlayer, it is arrested. Then, the load increases from $F \approx 9$ N to $F \approx 10$ N. During this increase, some amount of interfacial fracture occurs (see Fig. 9d) but the crack does not grow into the second layer. Thus, the increase in load corresponds to the crack being arrested by the weak interface. Finally, the crack re-nucleates in the second layer, resulting in the second load drop from $F \approx 10$ N to $F \approx 5$ N. After the second load drop, the crack continues to grow through the second layer until complete failure occurs.

We computed $\langle R \rangle^{(\text{arch})}/\langle R \rangle^{(\text{hom})}$ from the force-displacement response of the planar layered beam to be 1.18. This toughness enhancement is caused by the combined effects of two toughening mechanisms: interfacial fracture, and crack arrest and re-nucleation. By interfacial fracture we mean creation of additional fracture area, and by crack arrest and re-nucleation we mean the mechanism through which a crack is arrested at a weak interface between two layers and consumes additional energy to resume propagation in the adjacent, undamaged layer[11,12,56–58].

In the case of the cylindrical layered beam, we observed that the $F$-$w_0$ response is almost identical to that of its corresponding monolithic beam (see Fig. 9e). By examining the crack path for the cylindrical layered beam at different values of $w_0$ (e.g., see Fig. 9f) we observed that interfacial fracture occurs to roughly the same extent as in the planar layered beam. This results in a small toughness enhancement, $\langle R \rangle^{(\text{arch})}/\langle R \rangle^{(\text{hom})}$, of 1.05. However,

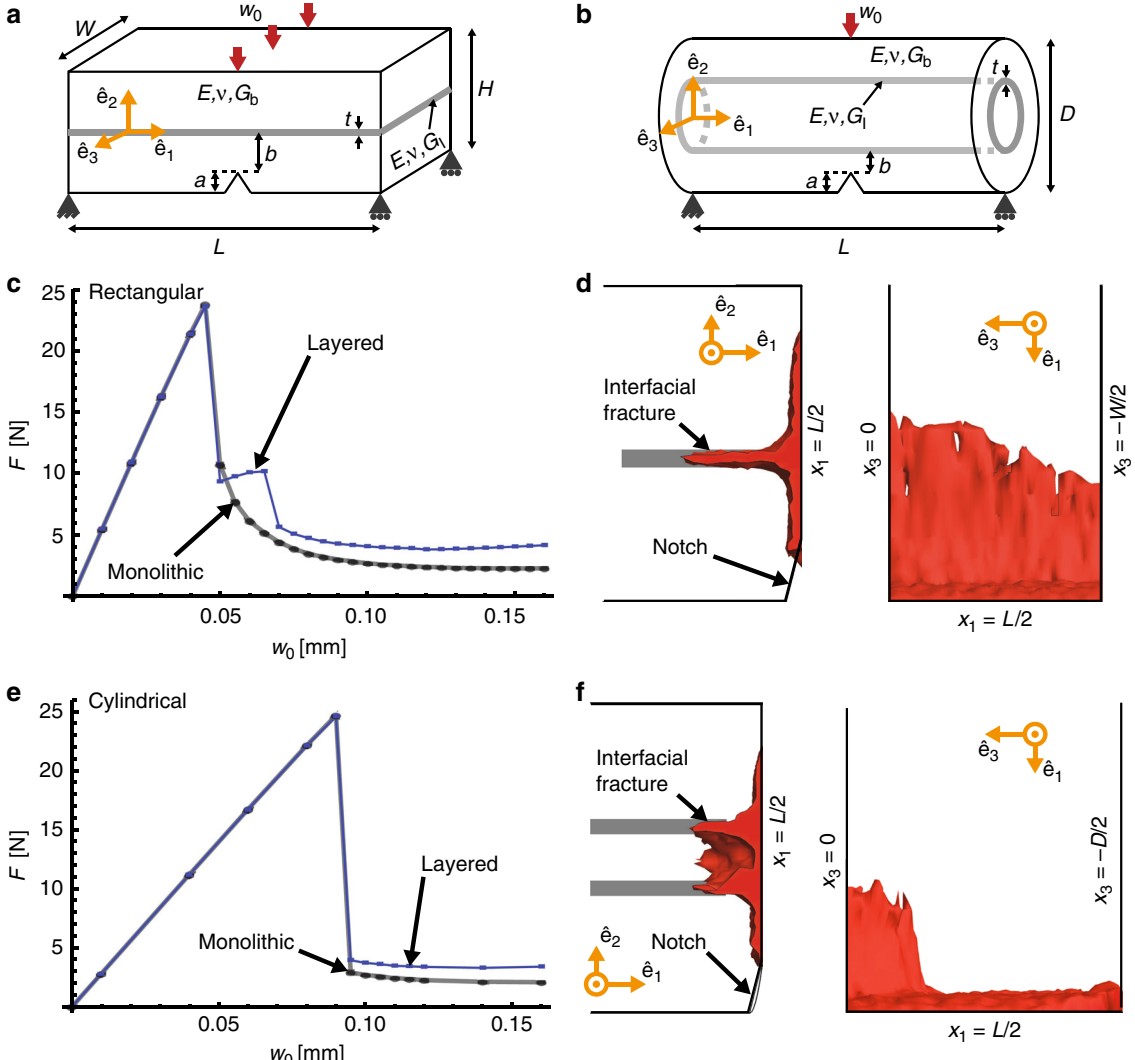

**Fig. 9 Regularized variational fracture model of beams with planar and cylindrical layered architectures. a** Geometry of the notched layered beam with one planar interlayer shown in gray. **b** Geometry of the notched cylindrical layered beam with one cylindrical interlayer shown in gray. **c** Load displacement ($F$-$w_0$) response of the planar layered beam loaded in three-point bending. **d** Crack paths in the planar layered beam as seen from the planes whose normal vectors are $\hat{e}_3$ and $\hat{e}_2$. The red region corresponds to the crack path predicted by the RVF calculations at a displacement $w_0 = 0.16$ mm. **e** $F$-$w_0$ response of the cylindrical layered beam loaded in three-point bending. **f** Crack paths in the cylindrical layered beam as seen from the planes whose normal vectors are $\hat{e}_3$ and $\hat{e}_2$. The red region corresponds to the crack path predicted by the RVF calculations at a displacement $w_0 = 0.16$ mm. The gray curves in **c** and **e** correspond to monolithic beams with the same geometry and bulk properties.

unlike the planar layered beam, the crack does not become arrested at the interlayer. The absence of the crack arrest and re-nucleation toughening mechanism explains why the toughness enhancement in the cylindrical layered beam is substantially lower than in the planar layered beam.

From this we conclude that while interfacial fracture does enhance the toughness of the cylindrically layered beam, its effect is small compared to the arrest and re-nucleation mechanism that occurs in the planar layered beam. This difference in toughening mechanisms could explain why the toughness enhancement is so much smaller for the *Ea.* spicules than for other SBMs with planar layered architectures.

The surprisingly limited toughness enhancement provided by the *Ea.* spicule's architecture reminds us that it is important to measure each SBM's toughness properties rather than categorizing it as tough solely based on the existence of a layered architecture. Furthermore, the contrast between our findings and previous speculations that the *Ea.* spicule's layers enhance their

toughness shows that the understanding of the relationship between layered architectures and toughness enhancement is not yet complete. By showing the extreme variability of the toughness enhancements that are provided by different layered architectures (see Section Comparison of toughness enhancements and Fig. 8), we hope to galvanize interest in developing a more complete understanding of this relationship. A better understanding of this structure-property relationship is crucial for developing useful bio-inspired designs and avoiding the pitfalls of naive biomimicry.

## Methods

**Spicule specimen preparation**. *Euplectella aspergillum* skeletons were received dried with the organic tissue removed (see Fig. 2a). We removed spicules from the basal portion of the skeleton using tweezers and cut ≈5 mm sections from roughly the midpoint along their length using a razor blade. *Tethya aurantia* spicules were received dried and separated from the sponge's organic tissue. We inspected the *Ea.* and *Ta.* spicules using a polarized light microscope and discarded specimens that were visibly cracked or damaged.

All specimens were stored in dry conditions prior to testing. We are aware that the mechanical properties of some SBMs can change substantially if they are soaked in water before testing. For example, the work of fracture of nacre that has been soaked in artificial seawater is 137% higher than that of the same nacre stored in dry conditions[11]. The soaking procedure is thought to restore the organic phases within them to their native, hydrated state. However, in Supplementary Methods, we compare the Young's modulus and bending failure strain of *Ea.* spicule specimens stored in wet and dry conditions and find no significant difference between the two (see Supplementary Methods for statistical analysis). Motivated by these results, we chose to test the spicules in their as-received, dry state.

**Spicule notching procedure**. We cut notches in the spicules using a focused ion beam (FIB). Before notching, we coated the spicules in 10 nm of carbon to prevent charge accumulation during the cutting procedure. We cut each notch in two steps. First, we used a relatively large accelerating current of 6.5 nA at 30 kV to make a broad cut (marked schematically in green in Fig. 4b). Then, we used a lower accelerating current of 460 pA at 30 kV to make a narrower cut (marked schematically in orange in Fig. 4b). The FIB was programmed to make the broad cut between 1.5 and 3 μm wide depending on the desired depth of the notch. Deeper notches required wider cuts in order to prevent the material that was ablated by the FIB from redepositing on the specimen. The narrow cut was programmed to be 250 nm wide. For specimens with very short notches (typically those for which $\alpha < 0.2$), the broad cut was omitted.

The actual widths of the cuts differed from these programmed values because the ion beam has a finite width. For each *Ea.* and *Ta.* spicule, we directly measured the widths of both the broad and narrow cuts from scanning electron micrographs (see e.g., Fig. 4c). Specifically, we measured the width of a cut at the points located closest to and furthest from the notch root. We then averaged the measurements taken at these two points and took this average value to be the width of the cut. Using this procedure we measured the broad cuts to be 3.58 ± 0.24 μm wide (mean ± standard error, $N = 35$; 24 of which were *Ea.* spicules and 11 of which were *Ta.* spicules) and the narrow cuts to be 501 ± 21 nm wide (mean ± standard error, $N = 61$; 35 of which were *Ea.* spicules and 26 of which were *Ta.* spicules).

This two step cutting process and the resulting notch geometry is similar to the procedure for preparing standard edge-notch bending specimens in which a notch is first cut using a diamond saw (broad cut) and then subsequently scored using a razor blade (narrow cut)[41,59,60].

Focused ion beams (FIBs) have previously been used to cut notches in micrometer-scale fracture specimens[31,40,61,62]. One concern about this technique is that the cutting mechanism—i.e., material ablation using gallium ions—can cause gallium ion implantation. This implantation alters local composition of the material and therefore could affect the local material properties near the notch root. Furthermore, ion implantation could result in the generation of compressive stresses within the material, which may have an effect on the measured fracture initiation toughness. A previous study[31] addressed these concerns by measuring the fracture initiation toughness of silicon (100) using FIB notched specimens and comparing these measurements to values obtained by other research groups that used macroscopic specimens and alternative notching procedures[63]. Their results also match those obtained from fracture tests performed on single crystal silicon specimens that were pre-cracked using a Knoop indenter and loaded in four-point bending[64]. Thus, they demonstrate that ion implantation does not appear to affect the measurement of fracture initiation toughness.

**Reporting summary**. Further information on research design is available in the Nature Research Reporting Summary linked to this article.

## Data availability
The data that support the findings of this study are available from the corresponding author upon reasonable request.

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

## Acknowledgements

We thank James Weaver for providing us with the *Ta.* spicules and Jarod Ferreira for his help constructing the mechanical testing device. The authors gratefully acknowledge support from the Office of Naval Research (Dr.Timothy Bentley) [Panther Program, grant number N000141812494]; the National Science Foundation [Mechanics of Materials and Structures Program, grant number 1562656]; the American Society of Mechanical Engineers [Haythornthwaite Research Initiation Grant]; and the NASA Rhode Island Space Grant Consortium.

## Author contributions

M.A.M. and S.K. carried out the experiments. K.V. carried out the computational fracture mechanics simulations. All authors discussed the results and wrote the paper. H.K. designed the research.

## Competing interests

The authors declare no competing interests.
