## [Peer Review File · Nature Communications]

Reviewers' Comments:

Reviewer #1:

Remarks to the Author:

The authors measure the fracture toughness of *E. aspergillum* spicules with a cylindrical layer structure of silica and protein and compare it to the properties of another biological spicule without such a layer structure. From their measurements, they conclude that the layered structure does not improve the fracture toughness.

Unfortunately, the paper contains a serious mistake which invalidates the conclusions of the paper: A layered structure improves fracture toughness and damage tolerance only, if an existing crack has to cross the layers, i.e. when it propagates perpendicular to the spatial variation of the material properties, see e.g. Fratzl et al. *Advanced Materials* 2007 and ref. [8]. The fracture mechanics specimens were fabricated by cutting a FIB notch into the spicules. Hereby, the notch length varies between 0.27 and 0.4 times the diameter of the spicule (Fig. 7). This means that the cut reaches the silica core of the spicule where no protein interlayers are present and the interlayers near the outside do not lie perpendicular to the crack growth direction, compare Fig. 1 to Figs. 3 and 6. Therefore, it is clear that the fracture resistance comes close to the fracture toughness of homogeneous calcite.

Additional notes:

(1) The increase of fracture toughness due to soft and/or compliant interlayers has been demonstrated experimentally in several papers, e.g. in Sistaninia et al., *Composite Structures* 2018.

(2) p.4, l.54: "Roughly speaking, fracture toughness—also known as crack growth resistance, R —is the amount of energy that a crack consumes to grow its area by a unit amount."

This is valid for brittle, homogeneous materials only, but not for inhomogeneous materials where a crack has to cross soft interlayers.

(3) p.9 Fig 5 and p.11: It is wrong to compute the fracture initiation toughness of the *E. aspergillum* spicule from Eq. (2). The change in potential energy and the crack extension during the pop-in only gives the crack growth resistance for the first crack growth increment after fracture initiation. See note (2).

Reviewer #2:

Remarks to the Author:

This paper sheds light onto previously alleged statements on the toughness enhancement caused by the layered architecture of *Euplectella aspergillum* (*Ea*) spicules. Through a combined approach of mechanical testing and computational mechanics, the authors unveil the reason for the low values of fracture initiation toughness and average crack growth resistance observed in their *Ea* spicules. This observation contrasts with previous results on *Ea* spicules, and it is misaligned with the high toughness values measured in other brittle biological composites, such as nacre and bone. The work is of rigor on both the experimental and computational aspects, and certainly holds broad implications not only because it resets the stage for a deeper understanding of the root causes behind the toughness mechanisms in layered architectures, but also because the insights here gained along with the proposed framework can serve as a springboard for the design of brittle yet tougher layered composites.

On the technical aspects, I appreciate the elucidation and clear distinction here put forward between the fracture initiation toughness and the average crack growth resistance along with their formulation used for the computational calculations. These are of high relevance for researchers working on this subject matter because they help quantify the toughness enhancement provided by a given architected material both at the instance of fracture initiation as well as during crack growth. These notions promote broad application to other stiff biological materials.

Below are constructive comments on a diverse range of points given to help improve the already

high quality of this work.

1) On the use of the variational fracture method (VFM) to discuss and explain the results. The set of past works from Ambrosio and Tortorelli, and later Francfort and Marigo (1998 & 2000) report numerical evidence for the variational fracture theory validation. While I certainly acknowledge the power and refinement of this method, I wonder whether after 18-20 years, the authors could have considered adding experimental validation. This would be certainly pertinent here considering the VFM is now applied to a biological layered material. Would for example, the fabrication and test of a simple synthetic analog, i.e. an artificial layered beam with controlled elastic properties for its bulk solid and its thin interlayer, be brought further evidence for the VFM accuracy and use in this work? Furthermore, given the main gist of the discussion lies on the insights gained from the computational results, are there any limitations to point out on the validity of application of the VFM?

2) The term "specific fracture initiation toughness", an important concept the authors build upon in this work, initially appears in the fourth line of the text, but its formal definition comes to light only at pag 4. I suggest providing an informal definition of this term right when it appears for the first time. It doesn't need to be technical (at pag 4 there is already one), yet a few words to clarify the reader of this journal of broad audience what the authors refer to would certainly be an asset.

3) Figure 3.

i) Fig 3A. although quite clear, I would suggest adding the standard dash – dot line on the specimen geometry in A and B to indicate exactly where the circular cross-sections below are taken.

ii) Fig 3C. References missing. I recommend including references in the caption for the data therein plotted.

iii) Fig 3A seems uncalled in the text. The first instance I see is only for Fig 3B and 3C at line 58.

4) Figure 8. It seems the y axis is on log scale. If this is the case, I suggest indicating it on the axis and/or in the caption. In addition, also Fig 8A is not referenced in the text, only Fig 8b is at line 240.

5) Figure 9. There seems to be an inconsistency with the choice of the view in Fig 9A and Fig 9B, the former displaying an axonometric view, whereas the latter is unclear (is it a front view?) because the cross-section on its right is a circle. How can this be? It should be an ellipse in an axonometric view, or I am missing something here. Another point on Fig 9F. Why at the notch location (bottom) a brown shaded zone appears? What is the physical meaning? Should the notch appear as open, as it is visible in the corresponding fig. 9D?

6) Statistics. This is a general comment that should be applied to relevant plots, table as well as text populated with data from this work or from the references. Statistical values are recommended for inclusion, where possible of course (if no data from the literature are available then the authors should nevertheless state this). The authors do a good job in providing mean values and standard deviations for the relevant mechanical property, e.g. line 173, but these values should be included also for the specimen geometry, so as to capture any discrepancy that might exist. For example, the start of the results section reports 25 Ea. and 12 Ta. Spicules were flexurally tested. But it seems no information (or I couldn't see) is given on the distribution of the geometric parameters that define these specimens. I would suggest including such information. I see much later in the text (pag 10) relevant values on the radius of curvature of the notch are given. Yet my comment here applies to the other main descriptors of the specimen geometry (cross-section radius, spicule length, layers thickness etc..). I would suggest including values of the specimen geometry up front in the results section, and use these raw values to build a short paragraph on the "specimen geometry characterization" showing any variations between data.

7) A similar recommendation applies to the plot in S1 where the effect of moisture on the mechanical properties is given for 33 dry Ea and 11 wet Ea spicules. I suggest including in the caption the statistical output of the experiments to corroborate the claim that no difference in Young's modulus and strain failure emerges from the testing results. On this point, I also note here that the scatter of the green and the light blue bars shows some differences. Can the authors expand on the reasons please?

8) Line 76. The sentence "orthonormal set of Cartesian basis vectors $\{e^1, e^2, e^3\}$, shown in Figure 4 (A), (B), and (D), and their corresponding Cartesian coordinates $\{x_1, x_2, x_3\}$ ". Since fig 4 reports only the basis vectors without any display of the system $\{x_1, x_2, x_3\}$, I would suggest to slightly amend the sentence as "orthonormal set of Cartesian basis vectors $\{e^1, e^2, e^3\}$ (Figure 4 (A), (B), and (D)), which correspond to the Cartesian coordinates $\{x_1, x_2, x_3\}$."

Minor typos

In the main text:

Line 276 layers, like that which appear >> like those?

In the supplementary information:

Line 14. That that > avoid duplicate

Line 39. compare > compared

Line 340. have been soaked > were soaked

Reference 2 in the supplementary information lacks some data, such as the year.

Reviewer #3:

Remarks to the Author:

Referee report:

To the manuscript entitled:

Architecture in Stiff Biological Materials: a Template for Toughness Enhancement, or a Siren Song?
by

Michael A. Monn, Kaushik Vijaykumar, and Haneesh Kesari

Indeed the sponge spicule model is exciting and it fascinated the scientists since 200 years. Now the authors gave an fairly extensive review on the studies reported until today by focusing on the toughness issues.

However what I criticize is that the selection of the papers they cite is very ambiguous.

The groups e.g. of Mayer, Fratzl, Weiner gave detailed report on this issue. Some pertinent report of these groups and also from others highlighted the importance of the proteins within the silica spicules and from the inter-layers. Also – importantly they mix spicules from demosponges with those from hexactinellides. This is not justified since the structure, composition of those spicules is different between these two taxa. I the authors would stick to one taxon – then they surely will come to another conclusion. Then the issue, basically two issues, become more clear.

I do not see any new ideas on the toughness of the layered mineralic biomaterials. Surely the mathematics behind those biomineral properties is apparently well done – even though I cannot verify the outlines to the details. This aspects seems to be new – but is by far too specific for a larger auditorium.

The authors gave a very attractive title for their manuscript – but they failed to give the readers an answer to: "...or a Siren Song". Just to mention that more conclusive results have to be worked out in the future is not sufficient.

Response to Reviewer #1's comments

1.1 “From their measurements, they conclude that the layered structure does not improve the fracture toughness.”

We would very humbly like to suggest that Reviewer #1 might have misunderstood the primary conclusion of our manuscript. We do not conclude, as Reviewer #1 appears to think, that “...the layered structure does not improve the fracture toughness.” Our primary conclusion is that due to its cylindrical nature, the *Ea. spicule*'s layered architecture provides relatively little toughness enhancement compared to other biological materials. We state this conclusion repeatedly throughout the manuscript, including in the abstract where we say “...spicule's layered architecture provides relatively little toughness enhancement,” and in the introduction where we state “We found that the toughness enhancement provided by the *Ea. spicule*'s architecture is much smaller than that provided by architectures seen in prototypically tough SBMs, like nacre...”

In the original version, we did state that the observed increase in initiation toughness $R(0)$ (but not average crack growth resistance $\langle R \rangle$) was not “significant” (lines 177–178 in the original version). However, we chose the adjective “significant” in adherence with Nature Publishing Group's guidelines for reporting statistical tests. It may have been a poor decision on our part to report the statistical test results on our $R(0)$ measurements, since those $R(0)$ measurements corresponded to different notch lengths and it is known, e.g. from the works of Gao *et al.* and Fratzl *et al.* [Gao, 1991, Fratzl *et al.*, 2007, Kolednik *et al.*, 2014, Kolednik *et al.*, 2011], that initiation toughness can change with notch length in materials with layered architectures. Therefore, it is possible that ultimately we are responsible for Reviewer #1's misunderstanding. We sincerely apologize for that misunderstanding.

We do, in fact, measure an increase in both fracture initiation toughness and average crack growth resistance. For example, we found the average enhancement in initiation toughness to be 1.07 ($N=17$). In the revised manuscript, which contains additional experiments, this enhancement is even larger—1.90 ($N=22$). Similarly, for our measurements of average crack growth resistance we found that the *Ea. spicule*'s architecture provided an average enhancement of 2.39 ($N=19$).

Our conclusion was that the observed enhancements were not as remarkable as those observed in prototypically tough biological materials. For example, the enhancement of initiation toughness and average crack growth resistance for nacre are, respectively, ≈ 196 and ≈ 150 (see Figure 8 (A), either in the original or revised manuscript).

We have consequently made two changes, which are listed as *Mc.2* and *Mc.3* in the LoC to clarify our findings. Through change *Mc.2* we have modified the abstract to include the magnitude of the toughness enhancements that we observed in the *Ea. spicules*. Through change *Mc.3* we have added a paragraph in the introduction to clarify that we did measure up to a ten fold increase in fracture initiation toughness but that this increase is small compared to the increases observed, for example, in nacre.

1.2 “Unfortunately, the paper contains a serious mistake which invalidates the conclusions of the paper”

The Reviewer points out that our original reported value for fracture initiation toughness enhancement of 1.07 was small due to the fact that most of our notches reached the spicule core where crack growth would not be perpendicular to protein interlayers. This comment is based on the assertion that a layered structure can only improve fracture toughness if an existing crack has to cross the layers. The Reviewer points us to the works of Fratzl *et al.* [Fratzl *et al.*, 2007], Kolednik *et al.* [Kolednik

et al., 2011], and Sistaninia *et al.* [Sistaninia et al., 2018] as support for this assertion.

We completely agree with the Reviewer's interpretation of our data. However, we do not understand why the notches being too long in our work to reveal the larger toughness enhancements predicted by the works of Fratzl *et al.* is a serious mistake. It would be a serious mistake only if the main conclusion of our work was that layered architecture does not enhance toughness. However, as we stated in our response to the previous comments, we do not conclude that the layered architecture does not enhance toughness in *Ea.* spicules. We only conclude that due to its cylindrical nature, the *Ea.* spicule's layered architecture provides relatively little toughness enhancement compared to other biological materials.

Even though it was not what we intended to say, the small enhancement in initiation toughness of 1.07 ($N=17$) reported in our original manuscript can be interpreted to mean that we are effectively saying that "the layered structure does not improve the fracture toughness." In this case, our work would appear at odds with the works of Fratzl *et al.* [Fratzl et al., 2007], Sistaninia *et al.* [Sistaninia et al., 2018], and Kolednik *et al.* [Kolednik et al., 2014, Kolednik et al., 2011] and the notches used in our experiments being too long to reveal the larger initiation toughness enhancements predicted by Fratzl *et al.* would indeed be a "serious mistake."

To prevent such interpretations, we have carried out additional experiments. These experiments were extremely challenging and took us close to half a year to complete. As the Reviewer anticipated, these new experiments reveal the larger initiation toughness enhancements predicted by Fratzl *et al.* Specifically, the additional experiments show that the enhancement in initiation toughness is 1.90 ($N=22$) on average (as opposed to 1.07 ($N=17$) in the original manuscript), and can be as large as 10. Thus, in the revised manuscript it is clear that the enhancement in initiation toughness due to the layered architecture is not negligible, is consistent with the predictions of Fratzl *et al.* [Fratzl et al., 2007], Sistaninia *et al.* [Sistaninia et al., 2018], and Kolednik *et al.* [Kolednik et al., 2014, Kolednik et al., 2011], and in fact provides strong experimental support to those works. The changes associated with these additional experiments are enumerated in *Mc.1* in the *LoC*. We also added a new paragraph to Section 2.3 in which we connect the finding of this work to the models for toughness enhancement presented by Fratzl *et al.*, Sistaninia *et al.*, and Kolednik *et al.*. This change is listed as *Mc.4* in the *LoC*.

1.3 "A layered structure improves fracture toughness and damage tolerance only, if an existing crack has to cross the layers, i.e. when it propagates perpendicular to the spatial variation of the material properties, see e.g. Fratzl et al. Advanced Materials 2007 and ref. [8]."

Assuming that the Reviewer means flat layers, we agree with the Reviewer's comment with regard to average crack growth resistance. In relation to initiation toughness, we also agree with the Reviewer's comment if the qualifier "only" is removed. That is, we are unable to agree that enhancement of initiation toughness is *only* possible if crack growth from a notch is perpendicular to the layers.

1.4 "Hereby, the notch length varies between 0.27 and 0.4 times the diameter of the spicule (Fig. 7). This means that the cut reaches the silica core of the spicule where no protein interlayers are present and the interlayers near the outside do not lie perpendicular to the crack growth direction, compare Fig. 1 to Figs. 3 and 6."

We agree with the spirit of the Reviewer's comment.

However, we would like to point out that the notch lengths in our original experiments varied between 0.18 and 0.64 times the diameter of the spicule (Figure 7 and Table S1 in the original submission). In the revised manuscript, which includes data from additional experiments, the notch lengths vary between 0.07 and 0.64 times the diameter of the spicule (Figure 7 and Table S2 in the revised

manuscript). From our previous work [Monn et al., 2015] we know that, roughly speaking, the notch tip would lie in the core when $\alpha > 0.3$, where α is the dimensionless notch length, a/D , and a and D are the notch length and the spicule's diameter, respectively.

Thus, in the original manuscript the notch tip does not reach the core in 29% of the initiation toughness experiments (see Table S1 in the original submission), and in the revised manuscript the notch tip does not reach the core in 45% of the initiation toughness experiments (see Table S1 in revised manuscript).

1.5 “Therefore, it is clear that the fracture resistance comes close to the fracture toughness of homogeneous calcite.”

We agree with the Reviewer's comment (assuming, of course, that the Reviewer meant silica when they wrote “calcite”).

1.6 “The increase of fracture toughness due to soft and/or compliant interlayers has been demonstrated experimentally in several papers, e.g. in Sistaninia et al., Composite Structures 2018.”

We thank the Reviewer for pointing us to the work of Sistaninia *et al.* Our work is not at odds with the results presented by Sistaninia *et al.* [Sistaninia et al., 2018] or the related works of Fratzl *et al.* [Fratzl et al., 2007] and Kolednik *et al.* [Kolednik et al., 2014, Kolednik et al., 2011]. In fact, we now use these works to explain the increase in initiation toughness with decreasing notch length that was revealed by our additional experiments in the revised manuscript. Furthermore, we thank Reviewer #1 in the acknowledgements section of the revised manuscript for pointing us to the works of Sistaninia *et al.*, Fratzl *et al.*, and Kolednik *et al.* and also for giving the explanation of why the initiation toughness for large notches is close to that of silica, which we have also included in the revised manuscript. See change *Mc.4* and *Mc.5* in the *LoC*.

1.7 “p.4, I.54: ‘Roughly speaking, fracture toughness—also known as crack growth resistance, R —is the amount of energy that a crack consumes to grow its area by a unit amount.’ This is valid for brittle, homogeneous materials only, but not for inhomogeneous materials where a crack has to cross soft interlayers.”

We respectively disagree with the Reviewer. See, for example, the works by Gao [Gao, 1991] and Hsueh *et al.* [Hsueh et al., 2018]. Consequently, we have not changed our statement that the Reviewer refers to in the revised manuscript.

1.8 “p.9 Fig 5 and p.11: It is wrong to compute the fracture initiation toughness of the *E. aspergillum* spicule from Eq. (2). The change in potential energy and the crack extension during the the pop-in only gives the crack growth resistance for the first crack growth increment after fracture initiation.”

We agree with second part of the Reviewer's comment, which is namely “The change in potential energy and the crack extension during the the pop-in only gives the crack growth resistance for the first crack growth increment after fracture initiation.” We further note that the crack growth resistance for the first crack growth increment is, by definition, the fracture initiation toughness.

However, we respectfully disagree with the first part of the Reviewer's comment, namely “It is wrong to compute the fracture initiation toughness of the *E. aspergillum* spicule from Eq. (2).”

The Reviewer might be conflating the different measures of fracture toughness, specifically initiation toughness with average crack growth resistance. Only the changes during the “first crack growth increment after fracture initiation” (by which we assume that the Reviewer means the limit of vanishingly small crack increments starting from when there is no crack, only notch) are needed for computing ini-

tiation toughness, but are insufficient for computing average crack growth resistance. Thus, Equation (2) is completely valid to compute initiation toughness.

To clarify, we derive Eqn. (2) directly from Griffith's theory of fracture, in which a necessary condition for a crack to start growing is that the energy release rate should equal or exceed the material's crack growth resistance. That is,

$$-\frac{d\Pi(\Delta a; w_s)}{d\Delta A(\Delta a; a, D)} = R(\Delta a) \quad (1)$$

where the function $\Pi(\cdot; w_s)$ is a function, which when supplied by the crack length Δa returns the system's potential energy $\Pi(\Delta a; w_s)$. The applied displacement w_s is a parameter in this function. On inputting the crack length Δa , the function $\Delta A(\cdot; a, D)$ returns the area of the fracture surface. This function is parameterized by the notch length a and the spicule diameter D . Equation (1) can be found in any basic textbook on fracture mechanics, such as [Gross and Seelig, 2017, Anderson, 2017, Kanninen and Popelar, 1985, Lawn, 1993].

The initiation toughness $R(0)$ is the value of the crack growth resistance $R(\cdot)$ in the limit of vanishing crack length. That is,

$$R(0) = \lim_{\Delta a \rightarrow 0^+} R(\Delta a). \quad (2)$$

Equation (2) in the manuscript follows from (1) and (2).

Response to Reviewer #2's comments

2.1 “On the use of the variational fracture method (VFM) to discuss and explain the results. The set of past works from Ambrosio and Tortorelli, and later Francfort and Marigo (1998 & 2000) report numerical evidence for the variational fracture theory validation. While I certainly acknowledge the power and refinement of this method, I wonder whether after 18-20 years, the authors could have considered adding experimental validation. This would be certainly pertinent here considering the VFM is now applied to a biological layered material.”

We agree with the Reviewer that experimental validation is extremely important for verifying the predictive capabilities of any computational tool. While we did not include this information in the manuscript, we have carried out extensive simulations to experimentally validate our finite element implementation of the Regularized Variational Fracture Theory (RVFT). We plan on presenting those validation results in a different manuscript that is currently under preparation². We briefly discuss three of our RVFT experimental validation results in the following paragraphs.

Experiment

Simulations

Figure 2. (a)–(b) Results from the bending test performed by Clegg *et al.* [Clegg *et al.*, 1990] on a SiC-graphite layered beam. (a) shows the failed configuration of the beam at the end of the bending test. (b) shows the load-displacement curve from the bending test. (c)–(d) Results from RVFT simulation of Clegg *et al.*'s bending test. The beam in the simulation was two dimensional, i.e., we assumed plane strain conditions. The SiC layers and the graphite interlayers (weak interfaces) are marked with symbols G_b and G_l , respectively, in (c). The span and width of the beam and the thicknesses of the SiC layers and interlayers are also shown marked in (c). (c) shows a representative crack pattern predicted by the RVFT simulations. The cracked region (where the damage field d is close to unity) is shown in red. The triangular notch is 0.1 mm at the base and 0.0625 mm in height. (d) shows the load-displacement curves from the RVFT simulations for different number of interlayers (weak interfaces). In the simulations toughnesses of the SiC layer and graphite interlayers were 500 J/m^2 and 50 J/m^2 , respectively, and the Young's modulus and Poisson's ratio of the SiC were 20.8 GPa and 0.3, respectively.

²“A variational mechanics theory for modeling the evolution of crack networks in composite materials with brittle interfaces,” Kaushik Vijaykumar, Pooya Yousefi, Christopher Larsen, and Haneesh Kesari. Under preparation for submission to the *Journal of the Mechanics and Physics of Solids*

Three point bending of a layered beam: Clegg *et al.* [Clegg *et al.*, 1990] created a layered beam by coating Silicon Carbide (SiC) sheets with graphite and then pressing and sintering them together. The graphite coating creates weak interfaces between the SiC sheets. This layered beam structure can be thought of as a 2D synthetic analogue for the *Ea.* spicules, where the SiC sheets and graphite coating represent the *Ea.* spicules' silica layers and protein interlayers, respectively. In fact, this structure is very similar to the “simple synthetic analogue”/“artificial layered beam” that Reviewer #2 mentions in their comment 2.2. Clegg *et al.* notched the SiC-graphite beam and loaded it until failure in a three-point bending configuration. (Note that we too notched the spicules and performed bending tests on them.) A failed configuration of the SiC-graphite layered beam from the bending test is shown in Figure 2 (a). The load-displacement curve from the bending test is shown in Figure 2 (b).

We simulated Clegg *et al.*'s bending test using the RVFT. The results from our RVFT simulation of Clegg *et al.*'s bending test are shown in Figure 2(c)–(d). The values for some of the parameters in the simulations were different from those in the experiments. For example, the values for the toughness and stiffness of SiC in our simulations (500 J/m^2 and 20.8 GPa , respectively) were different from those reported by Clegg *et al.* for the SiC in their experiments (28.25 J/m^2 and 450 GPa , respectively). We chose different values because the values for some of the parameters in Clegg *et al.*'s experiments were unavailable to us, such as the total number of SiC layers, thickness of a SiC layer (although this can be estimated from the images), the length and width/thickness of the SiC-graphite layered beam, toughness of the graphite interfaces, etc. Therefore instead of attempting to match our simulation results quantitatively with the experiments, we only attempted to match them qualitatively. So, we chose the parameter values in the simulations to make the RVFT's predictions qualitatively match experimental observations while ensuring that the RVFT computations did not become prohibitively expensive.

The RVFT is able to qualitatively capture many of the salient features of the experiment. For instance, as in the experiments in our RVFT simulations the crack begins to grow from the notch only when the energy release rate matches the toughness (critical energy release rate) of SiC. Also in agreement with the experiments, in our RVFT simulations the load drops after the first crack growth, the crack gets deflected on reaching the weak interfaces (interlayers), and following each deflection there is an increase in the load. Overall, the RVFT simulated crack has the same branched appearance as the experimentally observed crack (cf. Figure 2 (a) and (c)), and the RVFT simulated load displacement curve has the same stair-step appearance as the experimentally measured load-displacement curve (cf. Figure 2 (b) and (d)). Finally, the RVFT simulations are also able to capture the experimentally observed enhancement in the work of fracture provided by the layered architecture. Clegg *et al.* reported that the work of fracture for the SiC-graphite layered beam was 4625 J/m^2 , while that of monolithic SiC beam was 62 J/m^2 . That is, in the experiments the layered architecture enhances the work of fracture ≈ 75 times. In the RVFT simulations (containing three interlayers/weak-interfaces) the layered architecture enhances the work of fracture ≈ 2 times.

Asymmetric three point bending of a plate containing three holes Ingraffea and Grigoriu performed a asymmetric three-point bending test on a notched poly(methyl methacrylate) (PMMA) plate containing three holes [Ingraffea and Grigoriu, 1990] (see Figure 3 (a)–(b)). Here “asymmetric” refers to the fact that the notch was off center in the span direction (see Figure 3 (a)). The specimen's geometry and the experiment's loading are shown in Figure 3 (a). Figure 3 (b) shows the experimentally observed crack path. As can be noted, the crack emanates from the notch tip, curves to the right, and terminates in the middle hole.

We simulated Ingraffea and Grigoriu's experiment using the RVFT. In the simulations all dimensions

in the span and the width directions were the same as those in the experiments. The span and the width directions are, respectively, the \mathbf{e}_1 and \mathbf{e}_2 directions that are shown marked in Figure 3 (a). In the simulations we took the specimen to have infinite thickness, i.e., the RVFT simulations were two dimensional. Further details of our RVFT simulations are given in Figure 3's caption.

On comparing Figure 3 (b) and (c), visually, the crack path predicted by our RVFT simulation appears to match the experimentally observed crack path quite well. In order to quantify this match, for both the experiment and the simulation, we select a finite number of points on the crack path and measure the angles that the tangents to the crack path at those points make with the span direction (\mathbf{e}_1 direction in Figure 3 (a)). We show those angles as a function of the points' abscissa in Figure 3 (d). As can be seen, the crack path predicted by the RVFT indeed matches the experimentally observed crack path remarkably well.

Shear loading of a PMMA plate containing a vertical slit at its center Erdogan and Sih have reported fracture experiments on a PMMA plate that contained a vertical slit at its center [Erdogan and Sih, 1963]. The loading on the PMMA plate in the experiments is shown in Figure 4 (a). The crack path observed in the experiments is shown in Figure 4 (b).

We modeled Erdogan and Sih's experiment using the RVFT. We could not ascertain the dimensions of the PMMA plate in the experiments. Therefore, we chose to model only a small region around the central slit's top tip (region shown marked using dashed lines in Figure 4 (a).i) in our simulations. The modeled region was 2 mm in both the span and the width directions and contained a notch of 1 mm length on its bottom edge (see Figure 4 (b)). The notch was parallel to the width direction. The span and the width directions are, respectively, the \mathbf{e}_1 and \mathbf{e}_2 directions that are shown marked in Figure 4 (b). The loading on the modeled region in the simulations is also shown in Figure 4 (b). All other details in the simulations were the same as those in our simulation of Ingraffea and Grigoriu's experiment.

A representative result from our RVFT simulation is shown in Figure 4 (b). We measured the crack initiation angle in both the experiments and in our simulations. The crack initiation angle is the angle that a tangent to the crack at the point of its origin (which in the current context is the slit's top tip) makes with the width direction. The crack initiation angle in the experiments for the crack shown in Figure 4 (a).ii is $70 \pm 3^\circ$, while we found the crack initiation angle in our simulations to be $\approx 69.2^\circ$.

The regularized, variational fracture theory's predictions have been compared with experimental observations by other researchers as well, for example, see [Mesgarnejad et al., 2015], [Miehe et al., 2010], and [Wu et al., 2017].

The three experimental validation results that we just discussed in response to Reviewer #2's comment 2.1, and similar experimental validation results reported in literature make us feel confident that the RVFT is a dependable tool for gaining qualitative insight into fracture mechanics related phenomena. Needless to add, and as we will discuss in our response to Reviewer #2's comment 2.3, the RVFT does have several limitations with regard to its predictive capability, and requires further development before it is capable of making quantitative predictions. However, we have done our utmost to ensure that the issues that are known to limit RVFT's predictive capability were absent or negligible in the simulations that we report in the manuscript.

We did consider including the above discussion in the current manuscript in some form. However, we concluded that the current manuscript is perhaps not the appropriate forum for presenting our RVFT experimental validation results. Since presenting them in the supplementary information of the current

manuscript would not do them justice considering their significance. Presenting them in the main text would greatly change the scope of the manuscript, and make the manuscript unfocused and lengthy. Therefore, with regard to our RVFT experimental validation results we decided to limit ourselves to the above discussion for the current manuscript. However, as we mentioned earlier, we plan on presenting them along with all the requisite details and background in our future publication that will exclusively focus on RVFT².

2.2 “Would for example, the fabrication and test of a simple synthetic analog, i.e. an artificial layered beam with controlled elastic properties for its bulk solid and its thin interlayer, be brought further evidence for the VFM accuracy and use in this work?”

Experiment

Figure 3. (a) Geometry of the PMMA sample and the loading on it in Ingrassia and Grigoriu’s experiment. (b) The experimentally observed crack path in Ingrassia and Grigoriu’s experiment as reported by Bittencourt *et al.* [Bittencourt *et al.*, 1996] (c) shows the crack path predicted by our RVFT simulation of Ingrassia and Grigoriu’s experiment. The crack path (the region where the damage field $d \approx 1$) is shown in red. (d) A quantitative comparison of the RVFT predicted crack path with the experimentally observed one. See text for further details. In the simulations the toughness of PMMA was $\approx 394 \text{ J/m}^2$. Choi and Salem report toughness values for PMMA that, roughly, lie between 320 and 500 J/m^2 [Choi and Salem, 1993].

Absolutely. We too share the Reviewer’s interest in ascertaining the correctness/validity of RVFT by carrying out controlled experiments and comparing their results with the predictions from RVFT models of those experiments. The controlled experiments will involve synthetic analogues of *Ea* spicules in which, e.g., the mechanical properties of the constituent materials and interfaces will be well known. In fact, we have been trying to carry out such a comparison for the last two years. For example, Figure 5 shows some of our *Ea* spicule synthetic analogues, which we manufactured using laser cutting and 3D printing. Figure 5 (a) shows a two dimensional (2D) analogue that is very similar to the “artificial layered beam with controlled elastic properties for its bulk solid and its thin interlayer” that Reviewer #2 talks about in their above comment (comment 2.2). Figure 5 (b) shows a three dimensional (3D) analogue. Figure 5 (c) shows the mechanical testing stage that we designed and constructed for carrying out three-point bending fracture tests on our synthetic analogues. We discuss our 2D analogues further in the next paragraph.

We created our 2D analogues by laser cutting PMMA sheets and gluing them together using solvent based adhesives. Thus, in our 2D analogues the silica layers are represented by PMMA layers, and the protein interlayers (weak interfaces) are represented by the solvent based adhesive interlayers (PMMA-PMMA interfaces). Our aim is to measure the elastic properties and toughness of PMMA and the

Experiment

Simulation

Figure 4. (a.i) An illustration of the geometry and the loading conditions in Erdogan and Sih’s experiments [Erdogan and Sih, 1963]. (a.ii) A crack path from Erdogan and Sih’s experiments, as reported by Bittencourt *et al.* [Bittencourt *et al.*, 1996]. (b) Our RVFT model of Erdogan and Sih’s experiments and its predicted crack path.

toughness of the PMMA-PMMA interfaces in our three-point bend tests, input those properties into RVFT models of our experiments, and compare the predictions from those models for, e.g., the load-displacement curves and cracks paths with those from our experiments. This is ongoing work and will take one to two years.

We believe that the conclusion of our experiments on synthetic analogues is not necessary for the publication of our current results. This is for the following two reasons: (i) Firstly, as we discuss in our response to Reviewer #2's comment 2.1 we have already compared the predictions of RVFT with experimental observations from three-point bending fracture tests conducted on SiC-graphite layered beams, which were reported by Clegg *et al.* [Clegg *et al.*, 1990]. (Those beams are very similar to the "artificial layered beam" that Reviewer #2 mentions in their current comment (comment 2.2).) As we also discuss in our response to comment 2.1, we found from that comparison that RVFT is fully capable of making the type of qualitative predictions that we report in the manuscript. (ii) Secondly, though very valuable in ascertaining the validity of RVFT, we believe that our ongoing study involving synthetic analogues is outside the scope of the current manuscript. Our primary result—namely that the toughness enhancement provided by the layered architecture is greatly suppressed in *Ea. spicules*

Figure 5. Specimens under preparation for experimental validation of RVFT. (a) *Euplectella aspergillum*'s 2D synthetic analogues. Poly(methyl methacrylate)(PMMA) is laser-cut and glued back together to introduce weak interfaces perpendicular to the notch. (b) *Euplectella aspergillum*'s 3D synthetic analogue. A solid cylinder and a number of hollow cylinders of varying thicknesses and radii are 3D printed and assembled coaxially to resemble an *Ea.* spicule. (c) Fixture designed to conduct three-point bending test on the manufactured synthetic analogues.

and hence architectural motifs in stiff biological materials may not be worthy of emulation by default, and they may not be benefiting a particular organism in the same way that they benefit a different organism—does not critically hinge on the insight provided by the RVFT simulations.

We thank the Reviewer for pointing out a very interesting and important research direction.

2.3 “Furthermore, given the main gist of the discussion lies on the insights gained from the computational results, are there any limitations to point out on the validity of application of the VFM?”

Indeed, there are several issues that limit RVFT’s predictive capability. In our opinion, the most important of such issues is RVFT’s handling of compressive stresses.

A major issue in RVFT is that it does not distinguish between compressive and tensile stress states in dictating the nucleation and evolution of damage. This is because the evolution of d is primarily governed by the strain energy density field, Ψ_0 . The only information about the deformation that appears in the governing equation for d (Eqn. S25 in revised manuscript), is through Ψ_0 . Thus, in the RVFT for the purposes of dictating damage tensile and compressive stress states are indistinguishable when they correspond to the same strain energy density. Practically, what this means is that a crack can form even under purely compressive loading, which of course is absurd. This feature essentially renders the RVFT unusable in many important test cases. For example, consider the shear loading case shown in Figure 6. A plate with an edge notch is subjected to in-plane shear loading. The boundary condition and loading are detailed in Figure 6. As can be seen in subfigure (a), two crack branches emerge from the notch. The hoop stresses in the vicinity of the parent notch are compressive on the left side of the plate (see Figure 6 (c)). Therefore, the left crack branch is clearly unphysical; it is an anomaly of the RVFT.

Crack growth under compressive loading poses a major problem for the application of RVFT. However, a satisfactory solution to the problem of compressive failure is as yet unavailable. For example, in order to overcome the compressive failure problem Miehe et al. [Miehe et al., 2010] proposed to split the strain energy density into a “positive”, Ψ_0^+ , and a “negative”, Ψ_0^- , part. The positive and negative parts are, respectively, constructed using only the positive and negative eigenvalues of the strain tensor ϵ . The strain energy term in the RVFT functional (Eq. S24 in the revised manuscript) is changed to $\int_{\mathcal{B}} g(d)\Psi_0^+ + \Psi_0^- d\mathcal{B}$ so that now only Ψ_0^+ appears in the governing equation of d instead of Ψ_0 . The intuitive thinking underlying this strategy is that the tensile and compressive stress states will predominantly only affect Ψ_0^+ and Ψ_0^- , respectively. Therefore, a compressive loading will not appreciably change Ψ_0^+ and cause d to grow. The splitting strategy does help to some extent. For example, we repeated the shear loading test case using this strategy. The results are shown in Figure 6 (b). As can be see, the anomalous left branch now no longer appears.

However, the splitting approach is ad hoc, and lacks a rational mechanics justification. Most importantly, we found that the splitting strategy, in some cases, introduces new anomalies of its own. For example, consider the uniaxial tension test on a bar shown in Figure 6 (d) that we performed using the splitting strategy. As can be seen, damage has sporadically nucleated over the bar’s length. This damage pattern is spurious; an artifact of the splitting. Thus, in summary, the strategy of splitting the strain energy density into a positive and negative part works in certain cases while in other cases it very manifestly introduces new anomalies. A major point of concern therefore is the possibility that in some cases it may introduce anomalies that are not as easy to identify as those shown in Figure 6 (d) but which may lead to results that are significantly incorrect.

As we stated in our response to Reviewer comment 2.1, we have done our utmost to ensure that the above discussed and related issues that are known to limit RVFT’s predictive capability were absent

Figure 6. A limitation of RVFT: handling of compressive stresses. (a) shows the damage field on a notched square plate that is subjected to in-plane shear loading. Specifically, the plate's right boundary is encastered while a finite displacement is prescribed in the e_2 direction on the left boundary. The displacement in the e_1 direction on the left boundary is set to zero. Material property and geometry details are marked on the figure itself. The damage field shows a single crack in the right half of the plate. (b) shows the results for the same test case as in (a) expect that here the calculation was performed using the splitting strategy. The damage field now shows an additional, spurious branch on the left. (c) shows the hoop stresses, $\sigma_{\theta\theta}$, on a circle of radius 0.2 mm marked around the notch tip. The two different curves correspond to simulations whose corresponding damage fields are shown in (a) and (b). (d) shows the damage field for the case of a bar loaded in uniaxial tension. Symmetry boundary conditions are prescribed on the bottom and left boundaries of the bar. Displacements in the e_1 and e_2 directions on the right boundary are prescribed to be 0.18 and 0 mm, respectively. The bar has no notches and in its initial state has no damage. We solved this case using the splitting strategy, and as can be seen damage has sporadically localized along the bar's length.

or negligible in the simulations that we report in the manuscript.

In response to this comment we have made one change, which is listed as *Mc.6* in the *LoC*. Through change *Mc.6* we have added a new paragraph to Section S4.1 in which we describe this major limitation of RVFT (handling of compressive stresses).

2.4 “The term ‘specific fracture initiation toughness’, an important concept the authors build upon in this work, initially appears in the fourth line of the text, but its formal definition comes to light only at pag 4. I suggest providing an informal definition of this term right when it appears for the first time. It doesn’t need to be technical (at pag 4 there is already one), yet a few words to clarify the reader of this journal of broad audience what the authors refer to would certainly be an asset.”

We agree with the Reviewer that the concept of fracture toughness should be defined earlier in the manuscript. In response to this comment we have made one change, which is listed as *mc.2* in the *LoC*. Through change *mc.2* we have added a new sentence in the first paragraph of the introduction that provides a basic definition of fracture initiation toughness.

2.5 “Figure 3. i) Fig 3A. although quite clear, I would suggest adding the standard dash-dot line on the specimen geometry in A and B to indicate exactly where the circular cross-sections below are taken. ii) Fig 3C. References missing. I recommend including references in the caption for the data therein plotted. iii) Fig 3A seems uncalled in the text. The first instance I see is only

for Fig 3B and 3C at line 58.”

In response to this comment we have made one change, which is listed as *mc.3* in the *LoC*. Through change *mc.3* we have added the standard dash-dot line in subfigures 3A and 3B to indicate the location of the cross-sections. We have also added a reference to Figure 3A in the fourth paragraph of the introduction. Finally, we have identified the references from which the data in Figure 3C were obtained in the caption.

2.6 “Figure 8. It seems the y axis is on log scale. If this is the case, I suggest indicating it on the axis and/or in the caption. In addition, also Fig 8A is not referenced in the text, only Fig 8b is at line 240.”

In response to this comment we have made one change, which is listed as *mc.4* in the *LoC*. Through change *mc.4* we have modified the caption of Figure 8 to point out that the vertical axis has a logarithmic scale. We thank the Reviewer for this suggestion and think that pointing out that the vertical axis has a log scale reinforces the overall message of our work. We have also added a reference to Figure 8 (A) in the text of the revised manuscript (in the second paragraph of Section 2.5 of the revised manuscript).

2.7 “Figure 9. There seems to be an inconsistency with the choice of the view in Fig 9A and Fig 9B, the former displaying an axonometric view, whereas the latter is unclear (is it a front view?) because the cross-section on its right is a circle. How can this be? It should be an ellipse in an axonometric view, or I am missing something here. Another point on Fig 9F. Why at the notch location (bottom) a brown shaded zone appears? What is the physical meaning? Should the notch appear as open, as it is visible in the corresponding Fig. 9D?”

The Reviewer is not missing anything here. As the Reviewer has identified, we made some mistakes in preparing Figure 9 (B) and 9 (F). In Figure 9 (B), the cross-section on the right should not be rendered as a circle. In Figure 9 (F) the notch should appear open, like in Figure 9 (D). We have rectified our mistakes in the revised manuscript, see change listed as *mc.5* in the *LoC*. We thank the Reviewer for pointing out these mistakes.

2.8 “Statistics. This is a general comment that should be applied to relevant plots, table as well as text populated with data from this work or from the references. Statistical values are recommended for inclusion, where possible of course (if no data from the literature are available then the authors should nevertheless state this). The authors do a good job in providing mean values and standard deviations for the relevant mechanical property, e.g. line 173, but these values should be included also for the specimen geometry, so as to capture any discrepancy that might exist. For example, the start of the results section reports 25 Ea. and 12 Ta. Spicules were flexurally tested. But it seems no information (or I couldn't see) is given on the distribution of the geometric parameters that define these specimens. I would suggest including such information. I see much later in the text (pag 10) relevant values on the radius of curvature of the notch are given. Yet my comment here applies to the other main descriptors of the specimen geometry (cross-section radius, spicule length, layers thickness etc..). I would suggest including values of the specimen geometry up front in the results section, and use these raw values to build a short paragraph on the ‘specimen geometry characterization’ showing any variations between data.”

In response to this comment we have made three changes, which are listed as *Mc.7–Mc.9* in the *LoC*.

Through change *Mc.7* we have included a new table (Table 1 in the revised version) that provides statistical data for our measurements of diameter (D), span (L), and notch length (a) for both the

E. aspergillum and *T. aurantia* spicule specimens. Through change *Mc.8* we have added the standard error of the notch root radius measurements to paragraph 4 of Section 2.3. This completes the inclusion of statistical data for specimen geometry.

Through change *Mc.9* we have modified Table 2 in the revised version (Table 1 in the original version). This table contains the crack growth resistance data (both $R(0)$ and $\langle R \rangle$) used to compute the toughness metrics in Figure 8 for nacre, bone, antler, and conch shell. We have added statistical data such as mean, standard deviation, and sample number, when available. We have also clarified in the table footnotes when the number of specimens was not provided by a given reference or when the values in the table were estimated from graphs. We appreciate the Reviewer for pointing out this deficiency in our work.

2.9 “A similar recommendation applies to the plot in S1 where the effect of moisture on the mechanical properties is given for 33 dry Ea and 11 wet Ea spicules. I suggest including in the caption the statistical output of the experiments to corroborate the claim that no difference in Young’s modulus and strain failure emerges from the testing results. On this point, I also note here that the scatter of the green and the light blue bars shows some differences. Can the authors expand on the reasons please?”

In response to this comment we have made one change, which is listed as *mc.6* in the *LoC*. Through change *mc.6* we have added the mean and standard error for the Young’s modulus and bending failure strain of the wet and dry *E. aspergillum* spicules to the caption of Figure S1. The statistical treatment of this data is handled in more detail in Section S2, where we use both bias-corrected accelerated confidence intervals and two-sided *t*-tests to corroborate our claim that there is no significant difference in effective Young’s modulus or bending failure strain between the dry and wet spicules.

The difference in the scatter between the dry and wet spicules is most likely a consequence of the difference in sample number. That is, we only tested 11 spicules that were soaked in artificial seawater compared to the 33 dry spicules that were tested in [Monn and Kesari, 2017]. The effect of hydration on the mechanical behavior of the *E. aspergillum* spicules is certainly an important research topic, but a more in-depth treatment is beyond the scope of this work.

2.10 “Line 76. The sentence ‘orthonormal set of Cartesian basis vectors $\{\hat{e}^1, \hat{e}^2, \hat{e}^3\}$, shown in Figure 4 (A), (B), and (D), and their corresponding Cartesian coordinates $\{x_1, x_2, x_3\}$ ’. Since fig 4 reports only the basis vectors without any display of the system $\{x_1, x_2, x_3\}$, I would suggest to slightly amend the sentence as ‘orthonormal set of Cartesian basis vectors $\{\hat{e}^1, \hat{e}^2, \hat{e}^3\}$ (Figure 4 (A), (B), and (D)), which correspond to the Cartesian coordinates $\{x_1, x_2, x_3\}$ ’.”

In response to this comment we have made one change, which is listed as *mc.7* in the *LoC*. Through change *mc.7* we modified the sentence in their comment above in the way suggested by the Reviewer so that it now reads “The spicule specimen’s undeformed configuration can be described using the orthonormal set of Cartesian basis vectors $\{\hat{e}_1, \hat{e}_2, \hat{e}_3\}$ (Figure 4 (A), (B), and (D)), which correspond to the Cartesian coordinates $\{x_1, x_2, x_3\}$.”

2.11 “Minor typos In the main text: Line 276 layers, like that which appear » like those? In the supplementary information: Line 14. That that » avoid duplicate, Line 39. compare » compared, Line 340. have been soaked » were soaked, Reference 2 in the supplementary information lacks some data, such as the year.”

In response to this comment we have made one change, which is listed as *mc.8* in the *LoC*. Through change *mc.8* we have corrected the typos identified by the Reviewer. We thank the Reviewer for

identifying these small mistakes and allowing us to improve the quality of the manuscript.

Response to Reviewer #3's comments

3.1 “Now the authors gave an fairly extensive review on the studies reported until today by focusing on the toughness issues. However what I criticize is that the selection of the papers they cite is very ambiguous. The groups e.g. of Mayer, Fratzl, Weiner gave detailed report on this issue. Some pertinent report of these groups and also from others highlighted the importance of the proteins within the silica spicules and from the inter-layers.”

Since this work focuses on the toughness properties of the *Ea.* spicule, we focused our review on previous works that do the same. Many of these works have been published by Professor Mayer's group and are listed as references 5, 6, and 16 in the revised version of the manuscript. We do not agree that these references are ambiguous because, to our knowledge, they represent the main contributions to the study of the *Ea.* spicules from the perspective of toughness enhancement via experimental measurements. The toughness properties of spicules with layered architecture produced by other species of the class Hexactinellida have been explored in references 19–22 and 54 in the revised version of the manuscript.

We agree with the Reviewer that Professor Fratzl's group and his collaborators (such as Professor Kolednik) have also made significant contributions to this subject. Their works primarily focus on modeling the effects of an elastically heterogeneous, layered architecture on initiation toughness. They propose a crack tip shielding effect that is caused by the interaction between a crack and an elastic heterogeneity. Through change *Mc.4* listed in the *LoC* we have added an additional discussion of these works to Section 2.3 in the revised version of the manuscript. In this discussion, we describe the main premise of these models for toughness enhancement and point out how the analysis that we present in Section 2.5 constitutes a novel contribution by considering the cylindrical geometry of the *Ea.* spicule's architecture. We have included references to the contributions of these groups, which are listed as references 8, 25, and 45–47 in the revised version of the manuscript.

To our knowledge Professor Weiner's group has not published any works on siliceous spicules (i.e. Demosponge or Hexactinellid spicules). Consequently, we have not included a review of their works in this manuscript. However, we do not deny the importance of the protein scaffold within the spicule's silica nor the composition of the interlayers. Our review of this topic focuses on the contributions of Dr. Weaver and Professor Aizenberg which are listed as references 17, 18, 43, and 54 in the revised version of the manuscript. These works describe the composition and nanoscale structure of the spicules' silica. We revisit the importance of the protein scaffold and composition of the silica itself in the final paragraph of Section 2.5. In this paragraph we point out that we chose the *Ta.* spicules as a homogeneous control material based on the similarity of their composition and the composition of the *Ea.* spicules. We admit that the best choice for this homogeneous control material would be the core of the *Ea.* spicules themselves. However, we point out the difficulty of obtaining core samples large enough to test.

3.2 “Also—importantly they mix spicules from demosponges with those from hexactinellids. This is not justified since the structure, composition of those spicules is different between these two taxa.”

We agree that using spicules from the same taxonomic class would be the ideal choice. In fact, in Section 2.5 we state that “The *Ea.* spicules should instead be compared to a specimen composed of the same biogenic silica but which is monolithic. An ideal choice for this homogeneous control material would be a section of the solid silica core of the *Ea.* spicules. However, so far we have not successfully obtained a large enough section of the *Ea.* spicule core to perform fracture tests. Therefore, we chose

the *Ta.* spicules as what we believe to be the next best alternative.”

We used the *Ta.* spicules specifically as a material for comparison since they have been shown to have a similar volume-averaged bonding structure to the *Ea.* spicules [Weaver et al., 2010]. Specifically, ^{29}Si MAS NMR spectra of the *Ea.* and *Ta.* spicules indicate no distinguishable differences in the condensation of their silica. While this does not rule out differences in the proteinaceous scaffold, it does give a clear indication that the *Ta.* spicules are a close approximation in terms of silica composition. We have also measured the Young’s moduli of both the *Ea.* and *Ta.* spicules via three-point bending tests and found no significant difference between the two. This indicates that the elastic moduli of the silica from the *Ea.* and *Ta.* spicules are similar.

We agree with the Reviewer that differences in spiculogenesis between hexactinellids and demosponges will cause structural differences between the *Ea.* and *Ta.* spicules. We believe that this work constitutes a substantial improvement over past works focusing on toughness because it is the first to compare the *Ea.* spicules to spicules from a related sponge instead of to synthetic glass. We hope that this work will motivate the importance of continuing to seek better control materials for comparison, such as sections of the *Ea.* spicule core.

3.3 “I the authors would stick to one taxon—then they surely will come to another conclusion. Then the issue, basically two issues, become more clear.”

Regarding the conclusions of our work, we can assess the *Ea.* spicule’s toughness enhancement by analyzing the data we present for the *Ea.* spicules alone. Specifically, by taking the ratio of $\langle R \rangle$ to $R(0)$ for the *Ea.* spicules we can quantify the rise in their R curve during the propagation of a crack. This gives a rough sense of the effect of the *Ea.* spicule’s architecture on its toughness during crack propagation through what are considered “extrinsic” toughening mechanisms (see [Evans, 1990]). For the *Ea.* spicules, this ratio is approximately 22. We can compute this ratio for other biological materials using values found in Table 2. The values of $\langle R \rangle / R(0)$ for nacre, bone, antler, and conch are 3.5, 56.3, 140.0 and 1040.0, respectively. Thus, even without comparing to the *Ta.* spicules we can see that the increase in R provided by the *Ea.* spicule’s architecture is significantly smaller than most of these other biological materials.

We do not know what two issues the reviewer is referring to in their comment.

3.4 “I do not see any new ideas on the toughness of the layered mineralic biomaterials.”

The main conclusion and significance of our work is as follows. In the biomimetics and bioinspired engineering communities, the *Ea.* spicule’s layered architecture is being linked to potential toughness enhancement by default without direct experimental evidence to support that link [Mayer, 2011, Mayer, 2005, Kolednik et al., 2011, Walter et al., 2007]. Our primary conclusion is that due to its cylindrical nature, the *Ea.* spicule’s layered architecture provides relatively little toughness enhancement compared to other biological materials, such as nacre and conch shell.

Specifically, we found that in comparison to nacre, the initiation toughness and average crack growth resistance enhancements provided by the *Ea.* spicule’s layered architecture are smaller by a factor of 100 and 60, respectively. Thus, despite the layered architecture in *Ea.* spicules appearing very similar to those in nacre and conch shell—all three consist of micrometer thick ceramic/mineral layers glued together by nanometer-thin, soft, proteinaceous materials—and the *Ea.* spicules appearing to serve similar functions as nacre and conch shell—all three are stiff, marine, biological materials that serve structural functions in the organisms that create them—the layered architecture provides far less toughness enhancement for *Ea. spicules* than it does for nacre or conch shell. The stark contrast between the toughness enhancements observed in the *Ea.* spicules and in these other materials

demonstrates that the details of a material's layered architecture (e.g., flat vs cylindrical layers) can have extreme consequences on the toughening mechanisms that it enables and the resulting toughness enhancement it provides.

This is a very important finding, since it means that an architectural motif may not be benefiting a particular organism to the same degree that it does other organisms. In fact, the same architectural motif may be tied to different properties in different organisms, or not be tied to any property at all and just be a side effect of growth processes. The overarching implication is that the observation of an architectural motif's benefit in one biological material does not justify its use as a general template for reproducing that biological material's remarkable property enhancement in synthetic materials.

3.5 “The authors gave a very attractive title for their manuscript—but they failed to give the readers an answer to: “...or a Siren Song.” Just to mention that more conclusive results have to be worked out in the future is not sufficient.”

The title of the manuscript is meant to reflect the contrast between our findings and the widespread belief that biological materials with layered architectures have extraordinary fracture toughness. While we do state that “the contrast between our findings and previous speculations that the *Ea.* spicule's layers enhance their toughness shows that the understanding of the relationship between layered architectures and toughness enhancement is not yet complete”, we provide much more concrete conclusions as well. We enumerate some of the most important conclusions below.

1. The *Ea.* spicule's cylindrically layered architecture provides up to 10 fold increase in fracture initiation toughness. While this is certainly significant, it pales in comparison to the 200 fold increase observed in nacre (see Figure 8 (a)).
2. Perhaps more importantly, the increase in average crack growth resistance provided by the *Ea.* spicule's architecture is 60 times smaller than that provided by nacre's layered architecture.

We do not believe that the answer to the question posed in our title (i.e., are architectures in stiff biological materials a template for toughness or a siren song?) has a single, definitive answer. In conch shell, the crossed lamellar architecture clearly provides extraordinary enhancements to average crack growth resistance. In nacre, the brick and mortar architecture does the same for fracture initiation toughness. In the *Ea.* spicules, the cylindrically layered architecture only provides modest enhancements to both. While at first glance, all of these materials have similar architectures, however, the seemingly small architectural differences lead to quite different fracture toughness properties. Sometimes these properties are extraordinary, sometimes they are not. Addressing that subtlety is the main point of this work.

Bibliography

- [Anderson, 2017] Anderson, T. L. (2017). *Fracture Mechanics: Fundamentals and Applications*, pages 1–96. CRC press, Boca Raton, Florida, USA.
- [Bittencourt et al., 1996] Bittencourt, T., Wawrzynek, P., Ingraffea, A., and Sousa, J. (1996). Quasi-automatic simulation of crack propagation for 2d lefm problems. *Engineering Fracture Mechanics*, 55(2):321–334.
- [Choi and Salem, 1993] Choi, S. and Salem, J. (1993). Fracture toughness of pmma as measured with indentation cracks. *Journal of Materials research*, 8(12):3210–3217.
- [Clegg et al., 1990] Clegg, W., Kendall, K., Alford, N. M., Button, T., and Birchall, J. (1990). A simple way to make tough ceramics. *Nature*, 347(6292):455–457.
- [Erdogan and Sih, 1963] Erdogan, F. and Sih, G. (1963). On the crack extension in plates under plane loading and transverse shear. *Journal of basic engineering*, 85(4):519–525.
- [Evans, 1990] Evans, A. G. (1990). Perspective on the development of high-toughness ceramics. *Journal of the American Ceramic society*, 73(2):187–206.
- [Fratzl et al., 2007] Fratzl, P., Gupta, H. S., Fischer, F. D., and Kolednik, O. (2007). Hindered crack propagation in materials with periodically varying young’s modulus—lessons from biological materials. *Advanced Materials*, 19(18):2657–2661.
- [Gao, 1991] Gao, H. (1991). Fracture analysis of nonhomogeneous materials via a moduli-perturbation approach. *International Journal of Solids and Structures*, 27(13):1663–1682.
- [Gross and Seelig, 2017] Gross, D. and Seelig, T. (2017). *Fracture mechanics: with an introduction to micromechanics*. Springer.
- [Hsueh et al., 2018] Hsueh, C., Avellar, L., Bourdin, B., Ravichandran, G., and Bhattacharya, K. (2018). Stress fluctuation, crack renucleation and toughening in layered materials. *Journal of the Mechanics and Physics of Solids*, 120:68–78.
- [Ingraffea and Grigoriu, 1990] Ingraffea, A. R. and Grigoriu, M. (1990). Probabilistic fracture mechanics: A validation of predictive capability. Technical report, CORNELL UNIV ITHACA NY DEPT OF STRUCTURAL ENGINEERING.
- [Kanninen and Popelar, 1985] Kanninen, M. F. and Popelar, C. H. (1985). *Advanced fracture mechanics*. Number 15. Oxford University Press.
- [Kolednik et al., 2014] Kolednik, O., Predan, J., Fischer, F., and Fratzl, P. (2014). Improvements of strength and fracture resistance by spatial material property variations. *Acta Materialia*, 68:279–294.
- [Kolednik et al., 2011] Kolednik, O., Predan, J., Fischer, F. D., and Fratzl, P. (2011). Bioinspired design criteria for damage-resistant materials with periodically varying microstructure. *Advanced*

Functional Materials, 21(19):3634–3641.

- [Launey and Ritchie, 2009] Launey, M. E. and Ritchie, R. O. (2009). On the fracture toughness of advanced materials. *Advanced Materials*, 21(20):2103–2110.
- [Lawn, 1993] Lawn, B. (1993). *Fracture of brittle solids*. Cambridge university press.
- [Mayer, 2005] Mayer, G. (2005). Rigid biological systems as models for synthetic composites. *Science*, 310(5751):1144–1147.
- [Mayer, 2011] Mayer, G. (2011). New toughening concepts for ceramic composites from rigid natural materials. *Journal of the Mechanical Behavior of Biomedical Materials*, 4(5):670–681.
- [Mesgarnejad et al., 2015] Mesgarnejad, A., Bourdin, B., and Khonsari, M. (2015). Validation simulations for the variational approach to fracture. *Computer Methods in Applied Mechanics and Engineering*, 290:420–437.
- [Miehe et al., 2010] Miehe, C., Hofacker, M., and Welschinger, F. (2010). A phase field model for rate-independent crack propagation: Robust algorithmic implementation based on operator splits. *Computer Methods in Applied Mechanics and Engineering*, 199(45-48):2765–2778.
- [Monn and Kesari, 2017] Monn, M. A. and Kesari, H. (2017). Enhanced bending failure strain in biological glass fibers due to internal lamellar architecture. *Journal of the Mechanical Behavior of Biomedical Materials*, 76:69–75.
- [Monn et al., 2015] Monn, M. A., Weaver, J. C., Zhang, T., Aizenberg, J., and Kesari, H. (2015). New functional insights into the internal architecture of the laminated anchor spicules of euplectella aspergillum. *Proceedings of the National Academy of Sciences*, 112(16):4976–4981.
- [Ritchie, 2011] Ritchie, R. (2011). The conflicts between strength and toughness. *Nature Materials*, 10(11):817–822.
- [Sistaninia et al., 2018] Sistaninia, M., Kasberger, R., and Kolednik, O. (2018). To the design of highly fracture-resistant composites by the application of the yield stress inhomogeneity effect. *Composite Structures*, 185:113–122.
- [Walter et al., 2007] Walter, S., Flinn, B., and Mayer, G. (2007). Mechanisms of toughening of a natural rigid composite. *Materials Science and Engineering: C*, 27(3):570–574.
- [Weaver et al., 2010] Weaver, J. C., Milliron, G. W., Allen, P., Miserez, A., Rawal, A., Garay, J., Thurner, P. J., Seto, J., Mayzel, B., Friesen, L. J., Chmelka, B. F., Fratzl, P., Aizenberg, J., Dauphin, Y., Kisailus, D., and Morse, D. E. (2010). Unifying design strategies in demosponge and hexactinellid skeletal systems. *The Journal of Adhesion*, 86(1):72–95.
- [Wu et al., 2017] Wu, T., Carpiuc-Prisacari, A., Poncelet, M., and De Lorenzis, L. (2017). Phase-field simulation of interactive mixed-mode fracture tests on cement mortar with full-field displacement boundary conditions. *Engineering Fracture Mechanics*, 182:658–688.
- [Zhu and Joyce, 2012] Zhu, X.-K. and Joyce, J. A. (2012). Review of fracture toughness (g, k, j, ctod, ctoa) testing and standardization. *Engineering Fracture Mechanics*, 85:1–46.

Reviewers' Comments:

Reviewer #1:

Remarks to the Author:

The authors have significantly improved their paper, especially, by presenting additional experimental data describing the fracture behavior of spicules with small initial crack lengths.

There remains a single problem in the paper: The authors do not discriminate between fracture initiation toughness J_c and crack growth resistance R .

Everything is correct what the authors write in Point 1.8 in their Response to the Reviewer 1 comments, but it applies for homogeneous, elastic materials, only. Note that, for homogeneous elastic-plastic materials, the crack growth resistance R of a material can exceed J_c by several orders of magnitude. For inhomogeneous elastic materials, the crack driving force does not only depend on the crack length a and the load F , but is influenced by the material inhomogeneity. Then J_c can become different from R , if J_c is evaluated in the same way as for a homogeneous material.

The authors may or may not consider this point.

Reviewer #2:

Remarks to the Author:

In this thorough revision, the authors have brought forward additional analyses and clarifications on several points I had raised including the need to validate the Regularized Variational Fracture Theory. I appreciate the attempt to reproduce experimentally available data despite the unknown values of certain parameters. The qualitative agreement of the first example as well as the quantitative analysis comparing results with other papers are sufficient for the purpose, i.e. the RVFT is a dependable tool for gaining qualitative insight. I look forward to reading the paper under preparation for JMPS (I do agree on the choice to have a dedicated paper on this), where I suggest a quantitative agreement with controlled experiments and specimens with no dependence on other papers. This could add the quantitative character and provide accuracy values for use by the scientific community and beyond.

I appreciate also the detailed description of the limitations of the RVFT (on specimen under tension versus compression stress states) and the relevant paragraphs included in the revision. The addition of the statistics on all fronts provide the necessary context to the various sets of data populations therein included.

I thus support the publication of this article.

Reviewer #4:

Remarks to the Author:

After the detailed read of the revised manuscript as well as the rebuttal letter, I take the liberty to recommend this excellent piece of experimental work for publication after minor revision. The authors have "fought" exemplary against critical comments from the reviewers.

My remarks:

You wrote: "Understanding the link between layered architectures and toughness could help to identify new ways to improve the toughness of engineering composites".

This is a rather too strong statement, suggesting that the toughness of materials is basically linked only to the layered architecture.

Please note that the intercalation of the organics between silica layers is crucial for the mechanical properties of spicules. Comprehensive analysis of the interplay between organics and silica layered architecture will result in a more complete understanding of these structures and will lead to conscious biomimicry.

I recommend changing this sentence as follow: "Understanding the link between organic-inorganic layered architectures and toughness could help to identify new ways to improve the toughness of biomimetically engineering composites"

You wrote:Page 2 lines 18-20: Both the core and the layers are composed of silica and adjacent silica layers are separated by a thin (5–10 nm[17]) organic interlayer.

Here, it is strongly recommended to mention what kind of organic templates are intercalated between layered structures in spicules of glass sponges? It seems you have overlooked corresponding information listed below. This will help you as non-biologists to better understand the origin and localization of organic matrices within spicules.

Fig 3f, Fig 4f in: Ehrlich H., et al (2006) A modern approach to demineralisation of spicules in the glass sponges (Hexactinellida: Porifera) for the purpose of extraction and examination of the protein matrix. *Russian Journal of Marine Biology* 32(3):186–193

Fig.2A, Fig.3 and 5 in: Ehrlich H et al (2008) Nanostructural organisation of naturally occurring composites: Part I. Silica–collagen–based biocomposites. *Journal of Nanomaterials* 2008, Article ID 623838, 8 pages, doi: 10.1155/2008/670235)

Supplementary Fig 10 in: Ehrlich H., et al (2010) Mineralization of the meter–long biosilica structures of glass sponges is template on hydroxylated collagen. *Nature Chemistry* 2:1084–1088

Fig.2 + SI in: Ehrlich H et al (2016) Supercontinuum generation in naturally occurring glass sponges spicules. *Advanced Optical Materials* 4(10):1608–1613

Fig.14 and 25 in: Wysokowski M., Jesionowski T., Ehrlich H. (2018) Biosilica as source for inspiration in biological materials science. *American Mineralogist* 103(5):665–691

These papers must be discussed and cited in the final revision.

BROWN

Haneesh Kesari School of Engineering
Assistant Professor Brown University
of Engineering 184 Hope Street, B&H 612
Providence, RI 02912
Phone: 401-863-1418
Email: Haneesh_Kesari@brown.edu

Regarding: Point-by-point response to issues raised by referees (manuscript NCOMMS-18-11797493).

Dear Editors and Reviewers,

We thank you for your very valuable feedback. In response to your feedback, we have made the revisions listed in the section *List Of Changes (LoC)*, which can be found on page 2 of this response letter. We provide a point-by-point response to the Reviewers' comments in the following pages. Our responses to Reviewer #1's, Reviewer #2's, and Reviewer #4's comments can be found on pages 3, 4, and 5, respectively, of this response letter. Through these changes and our responses to your comments, we hope that we have addressed all of your concerns.

We thank you for your consideration of our revised manuscript.

Sincerely,

Michael A. Monn, Kaushik Vijaykumar, Sayaka Kochiyama, and Haneesh Kesari

List of Changes

Changes in response to Reviewer criticisms

- Mc.1* In response to Reviewer #4's comment (listed as comment 1.2 of this letter) regarding the sentence "Understanding the link between layered architectures and toughness could help to identify new ways to improve the toughness of engineering composites" in the abstract, we modified the sentence per the Reviewer's suggestion so that it now reads "Understanding the link between organic-inorganic layered architectures and toughness could help to identify new ways to improve the toughness of biomimetic engineering composites."
- Mc.2* In response to Reviewer #4's comment (listed as comment 1.3 of this letter) regarding the composition of the mineralized layers in the spicules, we have modified the second paragraph of the introduction so that it discusses the role of the organic matrix in silica mineralization and references past work that describes the composition of the silica and the origin of the organic matrix.

Response to Reviewer #1's comments

1.1 “The authors have significantly improved their paper, especially, by presenting additional experimental data describing the fracture behavior of spicules with small initial crack lengths. There remains a single problem in the paper: The authors do not discriminate between fracture initiation toughness J_c and crack growth resistance R . Everything is correct what the authors write in Point 1.8 in their Response to the Reviewer 1 comments, but it applies for homogeneous, elastic materials, only. Note that, for homogeneous elastic-plastic materials, the crack growth resistance R of a material can exceed J_c by several orders of magnitude. For inhomogeneous elastic materials, the crack driving force does not only depend on the crack length a and the load F , but is influenced by the material inhomogeneity. Then J_c can become different from R , if J_c is evaluated in the same way as for a homogeneous material. The authors may or may not consider this point.”

We are sincerely grateful to the Reviewer for giving our work another look.

We thank the Reviewer for pointing out the importance of including shorter notch lengths when measuring the initiation toughness of the *Ea.* spicules.

Response to Reviewer #2's comments

1.1 **"In this thorough revision, the authors have brought forward additional analyses and clarifications on several points I had raised including the need to validate the Regularized Variational Fracture Theory. I appreciate the attempt to reproduce experimentally available data despite the unknown values of certain parameters. The qualitative agreement of the first example as well as the quantitative analysis comparing results with other papers are sufficient for the purpose, i.e. the RVFT is a dependable tool for gaining qualitative insight. I look forward to reading the paper under preparation for JMPS (I do agree on the choice to have a dedicated paper on this), where I suggest a quantitative agreement with controlled experiments and specimens with no dependence on other papers. This could add the quantitative character and provide accuracy values for use by the scientific community and beyond. I appreciate also the detailed description of the limitations of the RVFT (on specimen under tension versus compression stress states) and the relevant paragraphs included in the revision. The addition of the statistics on all fronts provide the necessary context to the various sets of data populations therein included. I thus support the publication of this article."**

We thank the Reviewer for their in-depth review of our work. When preparing the JMPS paper containing details and validation of the RVFT method we will do our utmost to make a quantitative comparison "with controlled experiments and specimens with no dependence on other papers."

Response to Reviewer #4's comments

1.1 “After the detailed read of the revised manuscript as well as the rebuttal letter, I take the liberty to recommend this excellent piece of experimental work for publication after minor revision. The authors have “fought” exemplary against critical comments from the reviewers.”

We thank the Reviewer for their consideration of our manuscript, revisions, and appeal.

1.2 “You wrote: ‘Understanding the link between layered architectures and toughness could help to identify new ways to improve the toughness of engineering composites’. This is a rather too strong statement, suggesting that the toughness of materials is basically linked only to the layered architecture. Please note that the intercalation of the organics between silica layers is crucial for the mechanical properties of spicules. Comprehensive analysis of the interplay between organics and silica layered architecture will result in a more complete understanding of these structures and will lead to conscious biomimicry. I recommend changing this sentence as follow: ‘Understanding the link between organic-inorganic layered architectures and toughness could help to identify new ways to improve the toughness of biomimetically engineering composites.’”

We agree with the Reviewer. Therefore, in response to the above comment we have made one change, which is listed as *Mc.1* in the LoC. Through change *Mc.1* we modified the sentence in our abstract, quoted in the comment above, in the way suggested by the Reviewer so that it now reads “Understanding the link between organic-inorganic layered architectures and toughness could help to identify new ways to improve the toughness of biomimetic engineering composites.”

1.3 “You wrote: Page 2 lines 18–20: Both the core and the layers are composed of silica and adjacent silica layers are separated by a thin (5–10 nm [17]) organic interlayer. Here, it is strongly recommended to mention what kind of organic templates are intercalated between layered structures in spicules of glass sponges? It seems you have overlooked corresponding information listed below. This will help you as non-biologists to better understand the origin and localization of organic matrices within spicules. Fig 3f, Fig 4f in: Ehrlich H., et al (2006) A modern approach to demineralisation of spicules in the glass sponges (Hexactinellida: Porifera) for the purpose of extraction and examination of the protein matrix. Russian Journal of Marine Biology 32(3):186–193. Fig.2A, Fig.3 and 5 in: Ehrlich H et al (2008) Nanostructural organisation of naturally occurring composites: Part I. Silica-collagen-based biocomposites. Journal of Nanomaterials 2008, Article ID 623838, 8 pages, doi: 10.1155/2008/670235). Supplementary Fig 10 in: Ehrlich H., et al (2010) Mineralization of the meter-long biosilica structures of glass sponges is template on hydroxylated collagen. Nature Chemistry 2:1084–1088. Fig.2 + SI in: Ehrlich H et al (2016) Supercontinuum generation in naturally occurring glass sponges spicules. Advanced Optical Materials 4(10):1608–1613. Fig.14 and 25 in: Wysokowski M., Jesionowski T., Ehrlich H. (2018) Biosilica as source for inspiration in biological materials science. American Mineralogist 103(5):665–691. These papers must be discussed and cited in the final revision.”

We thank the Reviewer for pointing us to this important and relevant body of literature. We now discuss and cite the papers mentioned by the Reviewer in the revised manuscript.

Specifically, we have modified the second paragraph of the introduction to include greater detail on the type of organic templates found in the layers of glass sponges. This change is listed as *Mc.2* in the LoC. Through change *Mc.2*, we discuss and cite the above mentioned works as they relate to the composition of the silica layers and the function of the organic matrix as a template for silica mineralization. The modified part of the second paragraph in the introduction in the final revised

manuscript is as follows:

“Images of spicules from other Hexactinellid species that are partially dissolved in alkali solution reveal that the silica layers also contain a fibrillar organic matrix similar to the interlayers [Ehrlich et al., 2006, Ehrlich et al., 2008, Ehrlich et al., 2010]. Thus, this organic matrix serves both as a scaffold within the layers and a glue between them [Ehrlich et al., 2016]. It is believed that this organic matrix acts as a template for cell-assisted silica mineralization during the spicule’s growth process [Ehrlich et al., 2006, Ehrlich et al., 2008, Ehrlich et al., 2010, Wysokowski et al., 2018]. However, little is known about the growth process of *Ea.* spicules [Wysokowski et al., 2018].”

Bibliography

- [Ehrlich et al., 2010] Ehrlich, H., Deutzmann, R., Brunner, E., Cappellini, E., Koon, H., Solazzo, C., Yang, Y., Ashford, D., Thomas-Oates, J., Lubeck, M., et al. (2010). Mineralization of the metre-long biosilica structures of glass sponges is templated on hydroxylated collagen. *Nature chemistry*, 2(12):1084–1088.
- [Ehrlich et al., 2006] Ehrlich, H., Ereskovskii, A., Drozdov, A., Krylova, D., Hanke, T., Meissner, H., Heinemann, S., and Worch, H. (2006). A modern approach to demineralization of spicules in glass sponges (porifera: Hexactinellida) for the purpose of extraction and examination of the protein matrix. *Russian Journal of Marine Biology*, 32(3):186–193.
- [Ehrlich et al., 2008] Ehrlich, H., Heinemann, S., Heinemann, C., Simon, P., Bazhenov, V. V., Shapkin, N. P., Born, R., Tabachnick, K. R., Hanke, T., and Worch, H. (2008). Nanostructural organization of naturally occurring composites-part i: Silica-collagen-based biocomposites. *Journal of Nanomaterials*, 2008:53.
- [Ehrlich et al., 2016] Ehrlich, H., Maldonado, M., Parker, A. R., Kulchin, Y. N., Schilling, J., Köhler, B., Skrzypczak, U., Simon, P., Reiswig, H. M., Tsurkan, M. V., et al. (2016). Supercontinuum generation in naturally occurring glass sponges spicules. *Advanced Optical Materials*, 4(10):1608–1613. DOI: 10.1002/adom.201600454.
- [Wysocki et al., 2018] Wysocki, M., Jesionowski, T., and Ehrlich, H. (2018). Biosilica as a source for inspiration in biological materials science. *American Mineralogist: Journal of Earth and Planetary Materials*, 103(5):665–691.